# The endoplasmic reticulum connects to the nucleus by constricted junctions that mature after mitosis

Helena Bragulat-Teixidor [ID] [1,2,3]✉, Keisuke Ishihara [ID] [4], Gréta Martina Szücs [ID] [1,2] & Shotaro Otsuka [ID] [1,2]✉

## Abstract

**Junctions between the endoplasmic reticulum (ER) and the outer membrane of the nuclear envelope (NE) physically connect both organelles. These ER–NE junctions are essential for supplying the NE with lipids and proteins synthesized in the ER. However, little is known about the structure of these ER–NE junctions. Here, we systematically study the ultrastructure of ER–NE junctions in cryo-fixed mammalian cells staged in anaphase, telophase, and interphase by correlating live cell imaging with three-dimensional electron microscopy. Our results show that ER–NE junctions in interphase cells have a pronounced hourglass shape with a constricted neck of 7–20 nm width. This morphology is significantly distinct from that of junctions within the ER network, and their morphology emerges as early as telophase. The highly constricted ER–NE junctions are seen in several mammalian cell types, but not in budding yeast. We speculate that the unique and highly constricted ER–NE junctions are regulated via novel mechanisms that contribute to ER-to-NE lipid and protein traffic in higher eukaryotes.**

**Keywords** Endoplasmic Reticulum; Nuclear Envelope; Open Mitosis; Membrane Contact Site; Correlative Light-electron Microscopy
**Subject Categories** Cell Cycle; Organelles

## Introduction

The endoplasmic reticulum (ER) of the eukaryotic cell is the major site of lipid and membrane protein synthesis, and it acts as a hub for intracellular communication. This organelle extends from the nuclear envelope (NE) as a continuous membranous organelle throughout the cytoplasm. The ER in the cytoplasm consists of a network of tubules and sheets interconnected by numerous junctions (referred to as ER–ER junctions) (Nixon-Abell et al, 2016; Takakura et al, 2017). The ER makes extensive contacts with other membrane-bound organelles including mitochondria, the Golgi apparatus, and endosomes. These contact sites play essential roles in protein homeostasis, lipid transfer, metabolite shuffling, ion exchange, and signal transduction (Rossini et al, 2021). In these contact sites, ER membranes are tethered to, but not fused with the membranes of other organelles. In contrast to these contact sites, the ER membrane contacts the nucleus by direct fusion to the outer nuclear membrane (ONM) of the NE (West et al, 2011). We refer to these fusion contacts as ER–NE junctions. Previous studies have shown that NE proteins can diffuse from the ER to the NE (Zuleger et al, 2011) and that NE proteins accumulate at the ER when their transport to the NE is blocked (Boni et al, 2015; Ungricht et al, 2015). Therefore, despite the presence of ribosomes at the ONM, the majority of NE proteins are expected to be synthesised on the ER and subsequently transported to the NE through ER–NE junctions. The supply of NE lipids and proteins depends on the proper communication by ER–NE junctions.

ER–NE junctions were first visualized by electron microscopy more than 60 years ago in chemically fixed cells of rat spleen and maize root (Watson, 1955; Whaley et al, 1960). In these cells, the ER–NE junctions had a non-constricted wide cone-shaped base that resembled the morphology of ER–ER junctions (50–100 nm). On the contrary, narrow and constricted ER–NE junctions (25–30 nm in wide) were sporadically observed in the late 1980s in high-pressure frozen plant root tip cells (Craig and Staehelin 1988; Staehelin, 1997). High-pressure freezing preserved intracellular membranes in a native state much better than chemical fixation (Dahl and Staehelin, 1989), but the finding of constricted ER–NE junctions in plant cells in the late 1980s was based on a few images of two-dimensional EM. The lack of volumetric high-resolution EM techniques at that time precluded a generalizable and systematic characterization of the three-dimensional (3D) ultrastructure of ER–NE junctions. In the early 2010s, ER–NE junctions were observed in 3D with EM tomography in high-pressure frozen yeast cells, and their morphology was reported to be similar to that of ER–ER junctions (West et al, 2011). Nowadays, ER–NE junctions in all eukaryotic cells are thought to exhibit the same morphology as ER–ER junctions. Nonetheless, studies on the native ultrastructure of ER–NE junctions in high-pressure frozen mammalian cells are missing. Further, it remains unknown how ER–NE junctions are established and maintained during the cell cycle of open mitotic cells in which the NE disassembles during mitosis and reforms from ER membranes at the end of mitosis (Ungricht and Kutay, 2017). A systematic ultrastructural analysis of ER–NE junctions in high-pressure frozen mammalian cells is required for a generalizable understanding of the 3D ultrastructure of ER–NE junctions and their biogenesis.

In this study, we combined live-cell imaging, high-pressure freezing, and quantitative 3D EM to study systematically the ultrastructure of ER–NE junctions in mammalian cells in their

[1]Max Perutz Labs, Vienna Biocenter Campus (VBC), Vienna, Austria. [2]Medical University of Vienna, Max Perutz Labs, Vienna, Austria. [3]Vienna BioCenter PhD Program, a Doctoral School of the University of Vienna and Medical University of Vienna, Vienna, Austria. [4]Department of Computational and Systems Biology, School of Medicine, University of Pittsburgh, Pittsburgh, PA, USA. ✉E-mail: helena.bragulatteixidor@meduniwien.ac.at; shotaro.otsuka@univie.ac.at

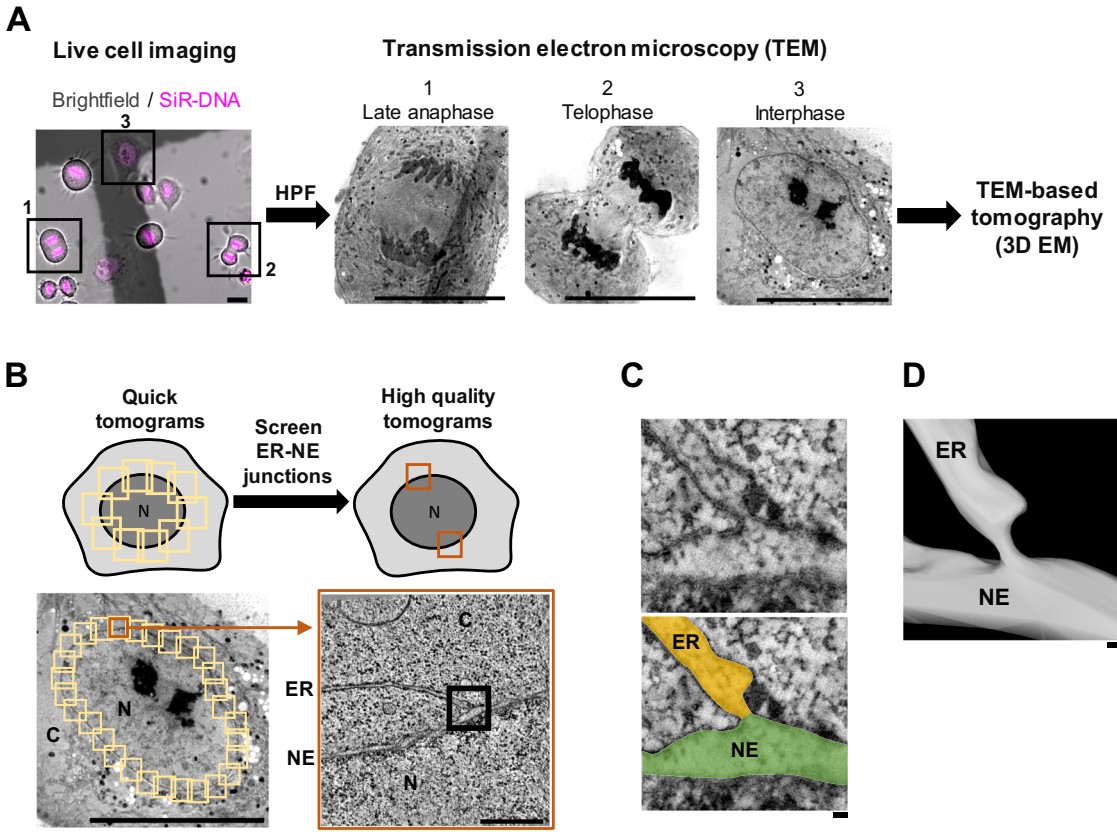

**Figure 1. Method to systematically study the 3D ultrastructure of ER–NE junctions along the cell cycle.**

(A) Correlative live cell imaging with electron microscopy to image the same cells using both imaging modalities. Cell-cycle progression of HeLa cells was monitored by live imaging in the bright-field and far-red (SiR-DNA) channels. Cells were high-pressure frozen (HPF) in late anaphase, early telophase or interphase, and relocated under a transmission electron microscope (TEM) for tomography. Scale bars: 20 μm. (B) Tomography-based screen to search for ER–NE junctions in an untargeted manner. First, 'quick' tomograms covering the entire NE in a section were acquired, and then high-quality tomograms were taken at regions with potential ER–NE junctions. Scale bars: 20 μm (overview), 500 nm (inset). (C) An electron tomographic slice of an ER–NE junction. The membrane of the ER and the outer nuclear membrane are clearly visible in the high-quality tomogram. In the bottom image, the ER and the NE are coloured in orange and green, respectively. Scale bars: 20 nm. (D) 3D mesh of the ER–NE junction shown in (C). Scale bar: 20 nm. ER endoplasmic reticulum, NE nuclear envelope, C cytoplasm, N nucleus. See also Movie EV1. Source data are available online for this figure.

native state. Strikingly, we discovered that ER–NE junctions have a constricted ultrastructure that is morphologically distinct from the junctions within the ER. By correlating live-cell imaging with electron tomography, we found that the ultrastructure of ER–NE junctions is remodelled during the cell cycle, and that the membrane constriction of ER–NE junctions starts in early telophase. These findings suggest that ER–NE junctions are regulated by a novel mechanism distinct from the one remodelling junctions within the ER. The highly curved and constricted ultrastructure of ER–NE junctions likely reflects their function as sites of lipid and protein traffic between the ER and nuclear membranes, which may regulate nuclear functions.

## Results

### An electron microscopy assay to study ER–NE junctions systematically throughout the cell cycle

To investigate the ultrastructure and abundance of ER–NE junctions at high temporal and spatial resolution throughout the cell cycle, we

established a workflow that combines live-cell imaging with high-resolution 3D EM (Fig. 1). Briefly, HeLa cells growing on carbon-coated sapphire discs were synchronized using a double thymidine block. Ten hours after the release from the thymidine block, most cells entered mitosis and we monitored their mitotic progression by time-lapse light microscopy. Of note, the double thymidine block does not affect the width of the ER and NE (details in Methods). When most dividing cells in the field of view reached the late anaphase or the telophase stage of mitosis, we rapidly froze the cells in a high-pressure freezing machine. Following freeze-substitution and infiltration in plastic resin, 250 nm sections were cut, and the cells identified by time-lapse imaging were imaged by transmission EM (Fig. 1A). Using this workflow, we can study the ultrastructure of cells and their organelles at precisely defined stages of the cell cycle. In this study, we focused on three time points: late anaphase (4–6 min after anaphase onset), early telophase (8–10 min after anaphase onset), and interphase (Fig. 1A).

We developed a two-step tomography-based screen consisting of 'quick' and 'high quality' tomograms to identify ER–NE junctions in the cells (Fig. 1B) (Bragulat-Teixidor and Otsuka, 2024). The quick tomograms covered the entire NE in a 250-nm-thick section at the middle plane of the nucleus. These quick

tomograms aimed to minimize the acquisition time of the tilt series while providing sufficient image quality to identify regions where the ER and the NE appeared in close proximity (Fig. 1B). At these regions with the ER proximal to the NE, high-quality tomograms were acquired to visualize the 3D ultrastructure of ER–NE junctions. The high-quality tomograms clearly resolved the lipid bilayer of the ER membrane continuous with the ONM of the NE (Fig. 1C,D, Movie EV1).

## ER–NE junctions are narrow and constricted in interphase

We characterized the ultrastructure of ER–NE junctions in interphase cells. High-quality tomograms revealed two distinct types of junction: those with a lumen (i.e. the ER/NE lumen was clearly visible at the interface) and those without a lumen (i.e. no lumen was visible at the interface, even though the membrane of the ER and the ONM appeared continuous) (Figs. 2A and EV1A, Movies EV2, EV3). We also saw examples in which the ER membrane was juxtaposed to the ONM or connected to it by filament-like densities (Fig. EV1A) that resemble the reported 'contact sites' between the ER and other intracellular organelles (Cai et al, 2022; Wozny et al, 2023). In this study, we focused on the ER–NE junctions in which the membranes of the ER and ONM appeared continuous.

To investigate in more detail the morphology of ER–NE junctions, we segmented the ER/NE membranes in tomographic slices to generate a 3D mesh model (Fig. 2B). The 3D meshes allowed us to see the sagittal (side view) and transversal (top view) planes of the junctions, and to measure the width, aspect ratio, and length of the junction neck (Figs. 2C and EV2A–C). The sagittal membrane profiles revealed that the ER–NE junctions with and without lumen both had an hourglass shape with a constricted neck ($n = 19$ ER–NE junctions from 9 cells) (Figs. 2D and EV1B). Top view profiles showed that the neck was $17.4 \pm 5.0$ nm for junctions with a lumen and $6.9 \pm 1.1$ nm for junctions without a lumen (Fig. 2E) (mean ± s.d., $n = 10$ junctions with a lumen, $n = 9$ junctions without a lumen). The width of junctions without a lumen corresponds to the thickness of a lipid bilayer, as the membrane profiles were drawn across the middle of the lipid bilayers. No ER–NE junctions larger than 24 nm in diameter were observed. The average length of the constricted neck was $10.4 \pm 5.4$ nm for junctions with a lumen and $4.9 \pm 1.5$ nm for junctions without a lumen (Fig. EV2D). The ER that connects to the constricted junction is either sheet-like, tubule-like, or with another morphology (Fig. EV1C), which indicates that ER–NE junctions are constricted irrespective of the shape of the approaching ER. Below the junctions with lumen, we noticed that the width of the perinuclear space was larger than in other regions of the NE (Fig. EV1D). In short, our ultrastructural analysis demonstrated that ER–NE junctions at the interface between the ER and the NE have a pronounced hourglass shape with a constriction of 7–20 nm wide and a length of 4–15 nm. Considering that our EM sample preparation protocol caused a 17% shrinkage of our specimen (Fig. EV3), the expected native width and length of ER–NE junctions are 8–24 nm and 5–18 nm, respectively.

To investigate whether the constricted morphology of ER–NE junctions is also seen in ER–ER junctions, we analysed the 3D morphology of ER–ER junctions in the same interphase cells that we screened for ER–NE junctions. We analysed specifically the ER–ER junctions that are located near the NE (Fig. EV1E) and that exhibited a similar topology to ER–NE junctions, i.e. an ER piece fused to the flat side of an ER sheet. In contrast to ER–NE

junctions, most ER–ER junctions had a wide cone-shaped base and much less marked membrane constrictions (Fig. 2F–H) ($n = 14$ ER–ER junctions from 4 cells). All the ER–ER junctions had a lumen. The mean width of ER–ER junctions was significantly greater and more variable than that of ER–NE junctions (Figs. 2I and EV2E) ($p$-value < 0.0001; Mann–Whitney test). In one example, we captured an ER piece forming simultaneously a constricted junction to the NE and a wide junction to an ER sheet (Fig. EV1F, Movie EV4), which highlights that the constricted hourglass shape is an exclusive feature to ER–NE junctions. Altogether, our ultrastructural analysis of interphase cells shows that ER–NE junctions have a highly curved and constricted hourglass shape that is distinct from broad perinuclear ER–ER junctions.

## ER–NE junctions become constricted in telophase

In metazoan cells that undergo open mitosis, the NE breaks down at the onset of mitosis and is absorbed in the ER (Ungricht and Kutay, 2017). The NE reforms again from membranes of the ER towards the end of mitosis in late anaphase and early telophase (Ungricht and Kutay, 2017). To investigate how and when the constricted morphology of ER–NE junctions begins to appear, we analysed the ultrastructure of ER–NE junctions when the NE is reforming. In late anaphase cells (4–6 min after the onset of anaphase), the ER and the NE are indistinguishable because the ER membranes just start contacting the surface of chromatin, presumably initiating NE reformation (Figs. 3A and EV4A, Movie EV5). At these ER membrane junctions touching chromatin (termed ER–ER/NE junctions), the ER pieces mainly connect to the edge of flat membrane sheets and not to the flat side of sheets. The junctions were not constricted and were significantly wider and more elongated than the ER–NE junctions in interphase (Figs. 3A–C and EV2E,F). Consequently, the junctions at the ER that contact the chromatin in late anaphase have not yet adopted the specialised morphology typical of ER–NE junctions in interphase.

We next looked in early telophase cells (8–10 min after the onset of anaphase). In telophase, the NE is clearly distinct from the ER because the membrane covers almost all the surface of the daughter nuclei (Figs. 3D and EV4B, Movie EV6). Ultrastructural analysis revealed that most ER–NE junctions in telophase started to become constricted (Fig. 3E). The average width of top view profiles was $26.2 \pm 7.9$ nm (mean ± s.d., $n = 18$ ER–NE junctions from 2 cells) (Fig. 3F), which is between ER–ER/NE junctions in anaphase and ER–NE junctions in interphase (Fig. EV2E). These data suggest that the constriction of ER–NE junctions starts at early cell-cycle stages (just 8–10 min after the onset of anaphase) and undergoes further maturation to form fully specialized junctions.

## The number of ER–NE junctions per cell increases slightly from telophase to interphase

To examine whether the abundance of ER–NE junctions changes from telophase to interphase, we quantified the NE surface area in the EM tomograms previously screened for junctions (Fig. 4A,B) in order to calculate the frequency of ER–NE junctions in telophase and interphase. In most junctions we could distinguish clearly the ER membrane continuous with the ONM (as in Figs. 2A and EV1A). A few junctions were categorized as ambiguous when the ER/NE membranes could not be traced with enough

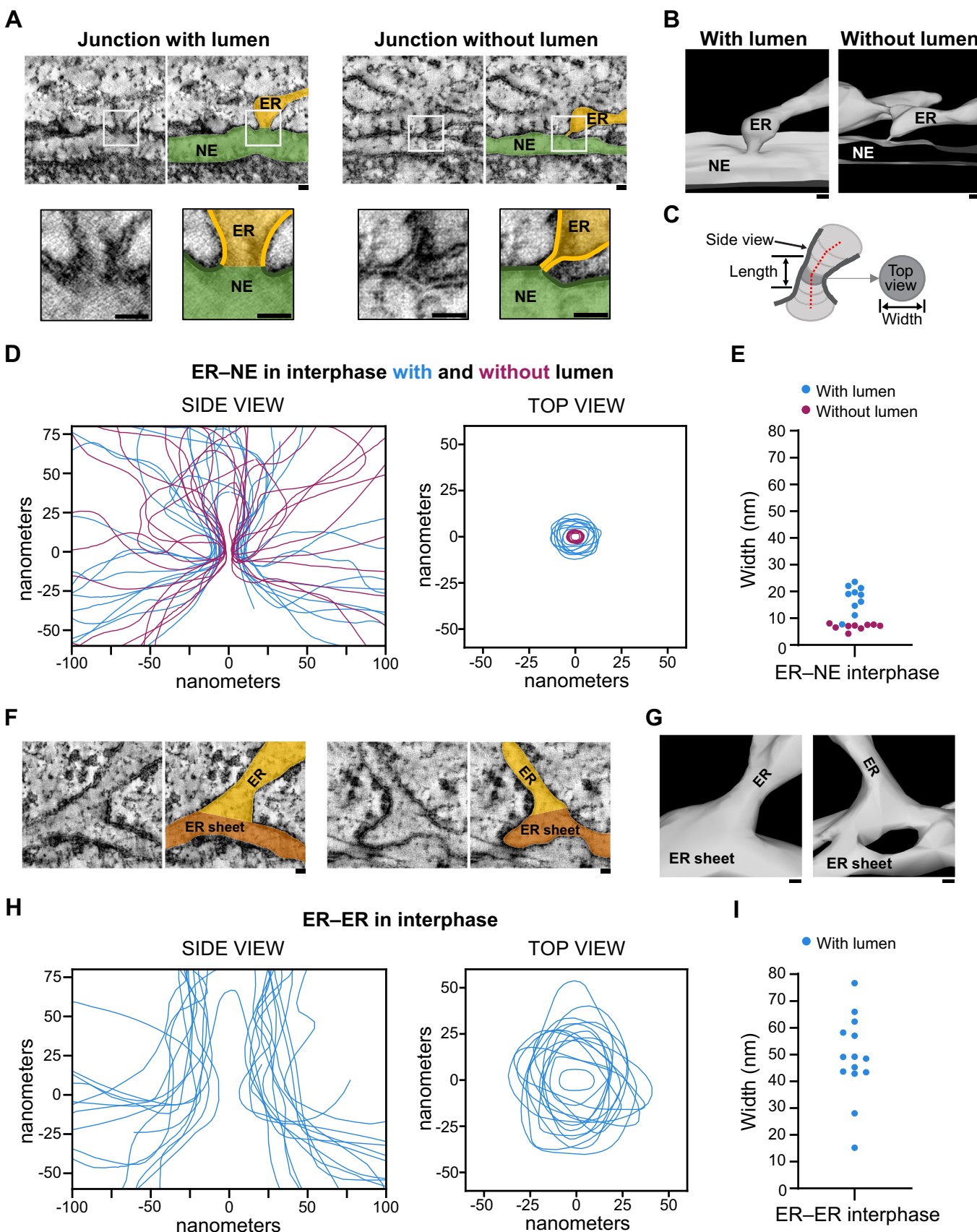

◄

**Figure 2.  The membranous junctions that link the ER to the NE in interphase are narrow, constricted, and distinct from the junctions within the ER.**

(A) Electron tomographic slices of ER–NE junctions with and without a lumen at their sagittal planes. For each junction, an enlargement of the region containing the junction is displayed below. The left image shows raw EM data; the right one, the EM data on which the ER and the NE are coloured in orange and green, respectively. Lipid bilayers are highlighted in bold lines in the insets. (B) 3D meshes of the ER–NE junctions shown in (A). See also Movie EV2. (C) The 3D mesh analysis allowed to obtain side and top view profiles, and the junction width, aspect ratio, and length. See also Fig. EV2A–C. (D) Membrane profiles of the side and top views of ER–NE junctions found in interphase cells. The profiles were drawn across the centre of the lipid bilayers. The side view profiles were overlaid vertically along the axis of the ER–NE junction. The top view profiles were aligned at their centroid. The profiles for the junctions with and without lumen are colour-coded in blue and magenta, respectively. $n = 19$ junctions from 9 cells from a single experiment. (E) Width of the constricted neck of ER–NE junctions in interphase. $n = 19$. (F) Electron tomographic slices of ER–ER junctions (in which ER tubules/sheets are fused to the flat side of ER sheets) at their sagittal planes. The left image shows raw EM data; the right one, the EM data on which the approaching ER piece and the flat ER sheet are coloured in orange and brown, respectively. (G) 3D meshes of the junctions shown in (F). (H) Membrane profiles of the side and top views of the ER junctions found in interphase cells. The side and top view profiles were overlaid as in (D). $n = 14$ junctions from 4 cells from a single experiment. (I) Width of ER–ER junctions. $n = 14$. Data Information: (A, B, F, G). Scale bars: 20 nm. Source data are available online for this figure.

confidence. These ambiguous junctions had only a small effect on our calculations because they were relatively few and similar in frequency in telophase and interphase cells (Fig. 4C). The frequency of all ER–NE junctions was ~0.18 μm$^{-2}$ in telophase (21 junctions in a screened NE surface area of 112 μm$^2$ from 2 cells) and ~0.13 μm$^{-2}$ in interphase (23 junctions in a screened NE surface area of 178 μm$^2$ from 9 cells) (Fig. 4C). These densities were much lower than those of junctions within the highly inter-connected ER network of sheets and tubules (Heinrich et al, 2021; Tikhomirova et al, 2022) or other NE-specific structures like nuclear pore complexes, which are 50–100 times more abundant (Otsuka et al, 2018).

We next asked if the total number of ER–NE junctions per nuclei increases in response to the nuclear growth from telophase to interphase. A previous study using 3D time-lapse fluorescence microscopy reported that HeLa cells have a total NE surface area of about 350–450 μm$^2$ in telophase (at 8–10 min after the onset of anaphase), and 700–900 μm$^2$ in interphase (Otsuka et al, 2016). Using these reported values of NE surface area and assuming a constant frequency of junctions around the whole nucleus, we estimated that a typical nucleus has approximately 70 junctions in telophase and 100 junctions in interphase (Fig. 4D). The number of junctions with a continuous lumen was similar in telophase and interphase cells, whereas the number of junctions without a lumen was greater in interphase (Fig. 4D). These data show that the number of ER–NE junctions increases slightly from telophase to interphase, although their abundance remains considerably lower than that of junctions within the ER network or nuclear pores at both cell cycle stages.

## The constricted morphology of ER–NE junctions is observed in different mammalian cells, but not in budding yeast

To determine if the constricted morphology of ER–NE junctions in interphase HeLa cells is common to other mammalian cells, we used whole-cell EM datasets publicly available in OpenOrganelle (Heinrich et al, 2021: Data ref: Heinrich et al, 2021; Xu et al, 2021: Data ref: Xu et al, 2021) to quantify ER–NE and ER–ER junctions in mouse pancreatic islet, HeLa, and human macrophage cells. In these datasets, cells were high-pressure frozen and freeze-substituted into plastic resin, and the entire cells were imaged by focused ion beam (FIB) scanning EM (FIB-SEM) at a voxel size of around 4 nm, which allowed us to investigate ER–NE junctions around the whole NE (Fig. EV5). Although the spatial resolution of

the FIB-SEM datasets is ten times lower than our EM tomograms, it was sufficient to identify and quantify the overall width of potential ER–NE junctions (Fig. 5A–E). To study ER–NE junctions in an unbiased manner, we took a stereology-based approach. The regions of interest were sampled in 1 μm radius spheres distributed uniformly over the entire nuclear surface, and those regions were inspected for potential ER–NE junctions (Fig. EV5). This unbiased inspection of the FIB-SEM datasets revealed that the width of ER–NE junctions in the pancreatic islet cells (Fig. 5A), HeLa (Fig. 5B), and macrophage (Fig. 5C) were significantly smaller than most ER–ER junctions (Fig. 5D,E), which is consistent with the ultrastructural observation in our high-quality EM tomograms (Fig. 2). Due to the limited spatial resolution of the FIB-SEM data, it is difficult to distinguish the ER–NE junctions with continuous membranes from the ER–NE 'contact sites' that we identified by tomography (Figs. 2A and EV1A). Therefore, the ER–NE junctions that we analysed in the FIB-SEM datasets may contain ER membranes that are in close proximity to, but not necessarily fused with, the ONM. Nevertheless, we rarely found ER–NE junctions that were not constricted. This analysis of FIB-SEM images confirms that ER–NE junctions are narrower than ER–ER junctions as seen in our EM tomograms of HeLa cells, and extends our findings to two other mammalian cell types.

To investigate whether hourglass-shaped ER–NE junctions are conserved among eukaryotes, we visualised ER–NE junctions in high-pressure frozen *S. cerevisiae* cells by tomography (Fig. 5F). In agreement with a previous study (West et al, 2011), we observed junctions between the NE and ER cisternae and tubules (Fig. 5F). Most ER–NE junctions had a wide cone-shaped base at the junction interface and no particular constriction (Fig. 5F). Quantification of the junction widths showed that ER–NE junctions in budding yeast are variable (Fig. 5G). We conclude that ER–NE junctions are generally not constricted in budding yeast; the highly constricted hourglass morphology of ER–NE junctions is a specialized feature of mammalian cells.

## Discussion

Whereas the molecular mechanisms that govern ER–organelle contact sites and their functions have been studied extensively (Rossini et al, 2021), the structure and the physiological role of the junctions connecting the ER to the NE remain poorly understood. Our study using correlative light and 3D electron microscopy reveals the native ultrastructure of ER–NE junctions in

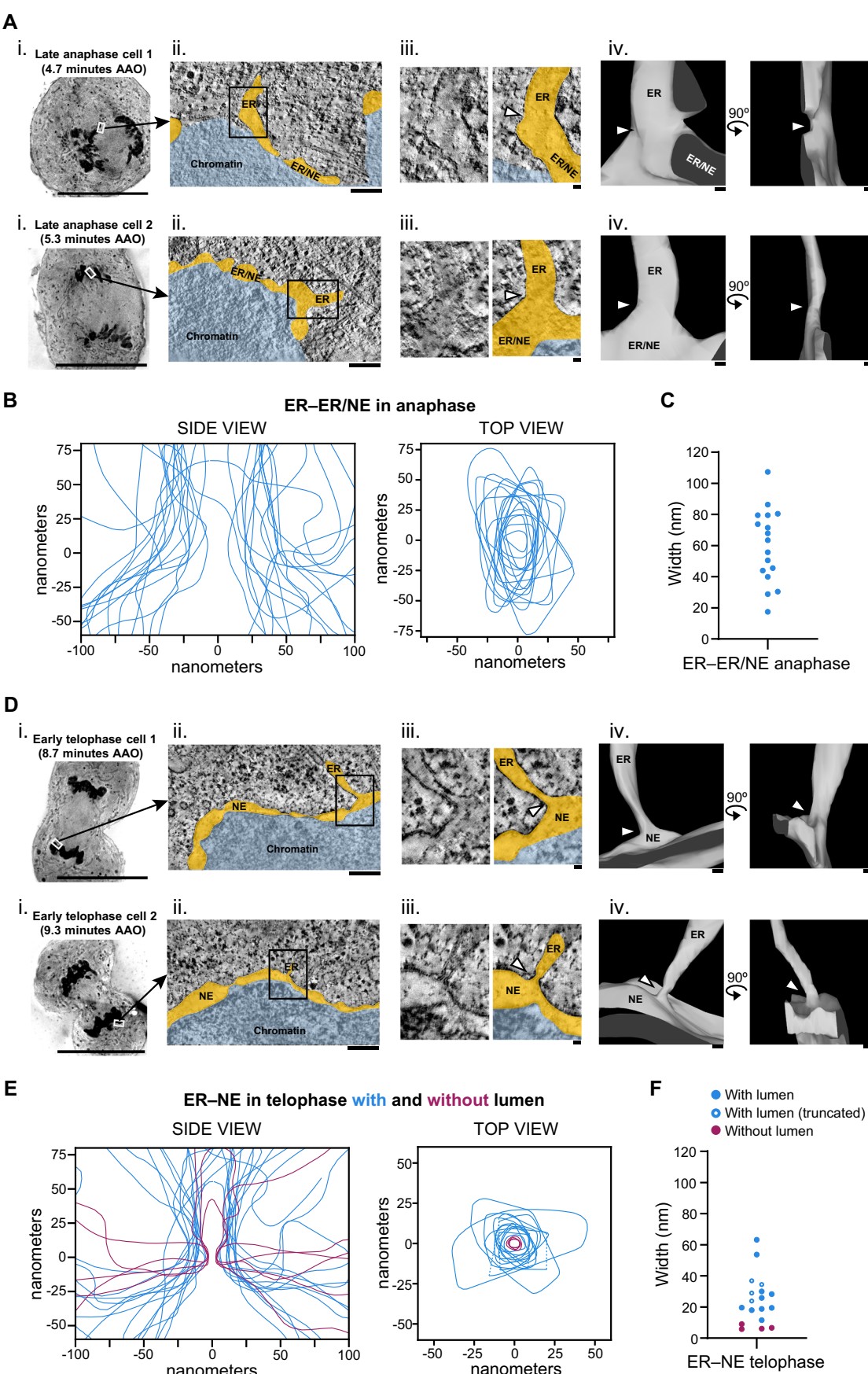

◄ **Figure 3. ER–NE junctions become progressively constricted during mitotic exit.**

(A) 3D ultrastructural analysis of ER–ER/NE junctions in late anaphase. (i) 2D EM micrographs of two different late anaphase cells. AAO: After Anaphase Onset. (ii) Tomographic slices of cells in late anaphase showing that the ER (orange) starts to contact the chromatin (blue), but the NE is still indistinguishable. (iii) Enlargement of the regions indicated in (ii). For each junction, the left image shows raw EM data; the right one, the EM data on which the ER and the chromatin are coloured in orange and blue, respectively. (iv) 3D meshes of the junctions and their 90°-rotated views. Junctions are indicated by white arrowheads in (iii) and (iv). See also Movie EV5. (B) Membrane profiles of the side and top views of the ER–ER/NE junctions found in late anaphase cells. $n = 17$ junctions from 3 cells from a single experiment. (C) Width of ER–ER/NE junctions in late anaphase. $n = 17$. (D) 3D ultrastructural analysis of ER–NE junctions in two different cells in early telophase. EM micrographs and the meshes are shown as in (A). See also Movie EV6. (E) Membrane profiles of the side and top views of the ER–NE junctions found in early telophase cells. The side and top view profiles are drawn as in Fig. 2B. $n = 18$ junctions from 2 cells from a single experiment. In 4 junctions, the entire top views could not be traced in the EM tomograms as they were truncated at the edge of the tomograms. For these 4 junctions, dashed lines indicate the truncated edge of the top view profiles. (F) Width of ER–NE junctions in early telophase. $n = 18$. Dots with a white filling indicate the 4 truncated top view profiles, whose width is underestimated. Data Information: Scale bars for (i): 20 μm; Scale bars for (ii): 200 nm; Scale bars for (iii, iv): 20 nm. Source data are available online for this figure.

high-pressure frozen mammalian cells at unprecedented spatial resolution. In interphase cells, the ER forms hourglass-shaped junctions with the ONM of the NE that are highly curved with a narrow constriction (7–20 nm in diameter). These ER–NE junctions are clearly distinct from broad perinuclear ER–ER junctions. The width of the NE below the junctions with lumen is larger than in other regions of the NE (Fig. EV1D), suggesting that LINC (linker of nucleoskeleton and cytoskeleton) complexes that typically maintain a NE width of 30–50 nm do not form below the junctions (Cain and Starr, 2015). During reassembly of the NE after anaphase onset, the junctions at the membranes contacting chromatin in late anaphase are large and elongated, and in early telophase they start to become constricted (Fig. 6A). This indicates that the formation of the hourglass shape of ER–NE junctions is initiated rapidly post anaphase and presumably maintained for the rest of the cell cycle. Our study is in agreement with the observations made by Craig and Staehelin over three decades ago that plant cells have constricted ER–NE junctions when they are high-pressure frozen (Craig and Staehelin, 1988). We show at high resolution and in 3D that the constricted morphology of ER–NE junctions is a common feature in mammalian cells. We also confirm a previous finding that the ER–NE junctions of budding yeast have a non-constricted shape (West et al, 2011). A cryo-EM tomogram of an ER–NE junction in *Chlamydomonas* also showed no constriction of the junction (Albert et al, 2017). Thus, our work suggests that the hourglass morphology of ER–NE junctions emerged only in higher eukaryotes, and evidence from several other species would be necessary to consolidate it.

What biophysical and molecular mechanisms might generate the high curvature of ER–NE junctions? Evidence from in vitro systems indicates that the generation and stabilization of highly curved membranes is energetically unfavourable in the absence of external forces (Lipowsky, 2022; Kozlov and Taraska, 2023). We hypothesize the existence of membrane remodelling proteins localizing at the neck of ER–NE junctions to maintain their constricted shape (Fig. 6B). Atlastins and Lunapark, which are known to form and stabilize three-way tubular junctions in the ER (Wang et al, 2016; Pawar et al, 2017), might be involved in generating and stabilizing ER–NE junctions. However, given that ER–NE junctions are clearly distinct from ER–ER junctions in morphology and abundance, additional proteins are presumably required to differentially remodel ER–NE junctions. While a dynamin mutant (K44A Dynamin 1) and ESCRT-III subunits CHMP1B and IST1 have been shown to create highly constricted membrane tubules, their inner diameters are 3.7 nm and 4.4 nm,

respectively (Antonny et al, 2016; Nguyen et al, 2020). Since the inner diameter of the ER–NE junctions without lumen is below 1 nm, the lumenless ER–NE junctions would be shaped by proteins other than dynamin or ESCRT. Our discovery that ER–NE junctions become constricted in telophase suggests that special remodelling proteins might be recruited to the NE as early as telophase. The proteins remodelling ER-NE junctions might be expressed predominantly or exclusively in higher eukaryotes, as ER–NE junctions are not highly constricted in yeast and *Chlamydomonas*. Unfortunately, our EM tomograms precluded the detection of consistent protein densities in the lumen or around the neck of ER–NE junctions that might correspond to large structural protein complexes. Future studies combining high-resolution imaging with molecular perturbations of the system will be necessary to identify potential regulators responsible for shaping and stabilizing ER–NE junctions.

How do ER–NE junctions form de novo during interphase? We found that the number of ER–NE junctions whose lumens are clearly connected does not change from telophase to interphase, whereas the number of junctions whose lumens do not connect increases 3-fold (Fig. 4D). Thus, we postulate two hypotheses for the biogenesis of ER–NE junctions. One hypothesis is that the junctions with a lumen are maintained throughout the cell cycle, whereas the junctions without a lumen arise de novo during interphase. The second hypothesis is that both types of ER–NE junctions are constantly remodelled (formed and degraded) during the cell cycle, like ER–ER junctions that form and break constantly during interphase (Nixon-Abell et al, 2016; Takakura et al, 2017). In either case, at least a small fraction of ER–NE junctions would form de novo from telophase to interphase. Regarding de novo formation of ER–NE junctions, we envision two scenarios. In one scenario, the NE would extend membrane protrusions into the cytoplasm. However, we observed no examples of protruding ONM in our EM images of interphase cells. Thus, we favour a second scenario in which ER–NE junctions are generated from ER tubules or sheets that approach the NE to contact and eventually fuse with the ONM. In our tomograms we found cases in which the ER membrane was juxtaposed (but not fused) to the ONM (Fig. EV1A). These contact sites might be assembly intermediates, and certain proteins may need to be recruited to mediate membrane fusion.

Given the constricted ultrastructure of ER–NE junctions, it is tempting to speculate that ER–NE junctions may act as a selective barrier for the transport of lipids, large protein complexes, and protein aggregates between the ER and the NE (Fig. 6B). A previous study showed that phosphatidylserine (one of the major acidic phospholipids

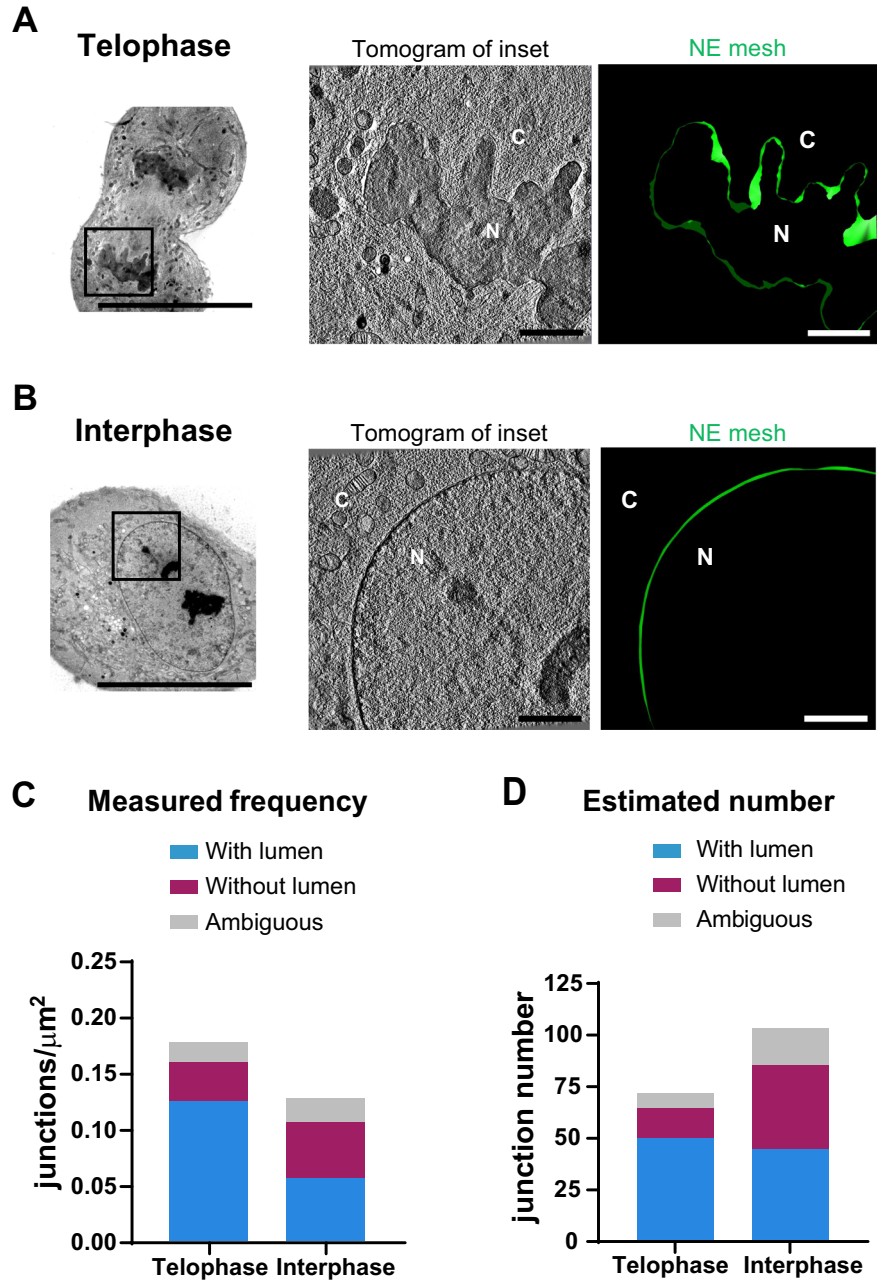

**Figure 4. The amount of ER–NE junctions changes slightly from early telophase to interphase.**

(A) 3D mesh of the NE in an EM tomogram for quantifying its surface area in an early telophase cell. Left: an overview 2D-EM micrograph; Middle: a tomographic slice of the region indicated in the overview image; Right: 3D mesh of the NE obtained from the tomogram shown in the middle panel. C: cytoplasm, N: nucleus. Scale bars for overview: 20 μm; for inset: 200 nm. (B) 3D mesh of the NE in an interphase cell. Images are shown as in (A). (C) Frequencies of the different types of ER–NE junctions in early telophase and in interphase cells calculated by dividing the number of junctions by the NE surface area. (D) Estimated total number of ER–NE junctions in average nuclei of early telophase and interphase cells, calculated by multiplying the frequencies in (C) with the total NE surface of early telophase and interphase nuclei measured in a previous study (400 μm² for early telophase and 800 μm² for interphase) (Otsuka et al, 2016). Source data are available online for this figure.

in eukaryotic cells) is over 10-fold enriched in the ER membrane when compared with the ONM despite the continuous nature of the ER and NE, and this phosphatidylserine enrichment occurs in mammalian cells and not in budding yeast (Tsuji et al, 2019). Since phosphatidylserine is a cylindrical-shaped lipid that favours the formation of flat membranes (Peeters et al, 2022), ER–NE junctions with highly-curved membranes may exclude phosphatidylserine and prevent its diffusion

into the ONM. As NE lipid composition affects NE protein stability (Lee et al, 2023), ER–NE junctions may thus play a key role in NE lipid/protein homeostasis, as proposed previously (Bahmanyar and Schlieker, 2020). The average inner diameter of ER–NE junctions with lumen is approximately 10 nm (after considering the shrinkage due to EM sample preparation), which is larger than the diameter of most proteins (~5 nm), but small enough to restrict the diffusion of large

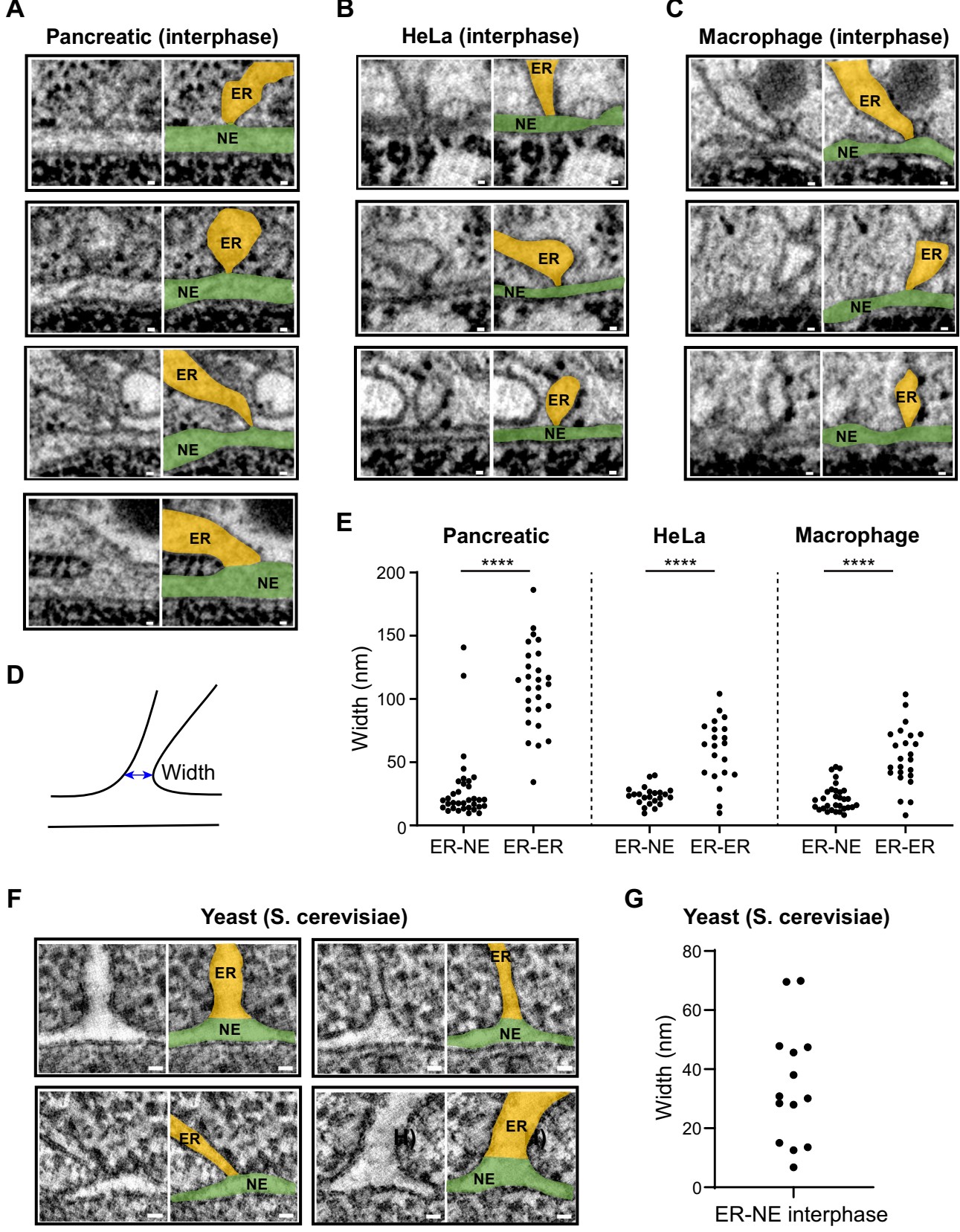

**Figure 5. ER–NE junctions are constricted in several mammalian cell lines, but not in budding yeast.**

(A–C) ER–NE junctions in FIB-SEM images of a mouse pancreatic islet cell (pancreatic) (A), a HeLa cell (B), and a human macrophage (C). For each junction, the left image shows raw EM data; the right one, the EM data on which the ER and the NE are coloured in orange and green, respectively. The images in the fourth row in (A) show one of the two junctions in the pancreatic cell with a width above 100 nm. (D) Scheme depicting how the width of junctions was measured using a 2-point line (blue). (E) Widths of ER–NE and ER–ER junctions in a mouse pancreatic islet cell, a HeLa cell, and a macrophage. n = 34 ER–NE and 27 ER–ER junctions for a mouse pancreatic islet cell, n = 23 ER–NE and 21 ER–ER junctions for a HeLa cell, and n = 31 ER–NE and 24 ER–ER junctions for a macrophage. ****p-value < 0.0001; two-tailed Mann–Whitney test. (F) Tomographic slices of ER–NE junctions of budding yeast (Saccharomyces cerevisiae) cells at their sagittal planes. (G) Widths of ER–NE junctions measured in the yeast cells. n = 14 junctions from 8 cells from a single experiment. Data Information: (A–C, F) Scale bars: 20 nm. Source data are available online for this figure.

**A**

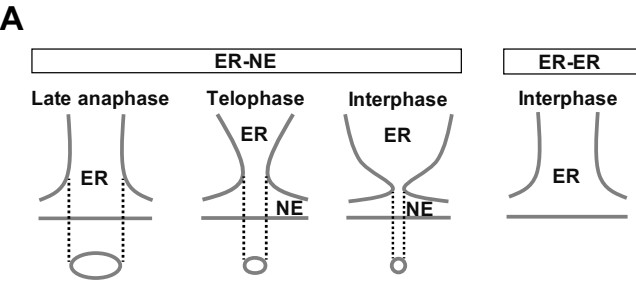

**B**

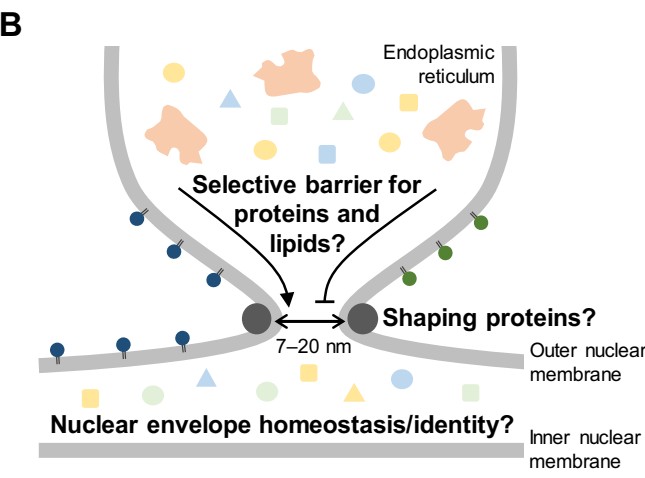

**Figure 6. Model for the formation and function of ER–NE junctions in higher eukaryotes.**

(A) ER–NE junctions mature progressively from late anaphase and early telophase into a constricted hourglass-shaped morphology in interphase that is distinct from the broad junctions within the ER. (B) Implications and potential functions associated with the constricted morphology of ER–NE junctions in interphase.

organelles such as mitochondria, endosome, and plasma membrane respond to various environmental cues including nutrient stress, ER stress and calcium stimulation (Donahue et al, 2022; Jang et al, 2022). Similarly, the membrane junctions between the ER and the NE might change their morphology in response to environmental stimuli. It would be interesting to examine the dynamic nature of the junctions under physiological stresses.

Newly synthesised NE proteins that cross ER–NE junctions play crucial roles in gene expression, nuclear organization, and nuclear pore biogenesis, as well as in development and disease (Gauthier and Comaills, 2021). ER–NE junctions may contribute to these key nuclear functions by acting as selective barriers for proteins and lipids. Dysregulated nuclear size, mislocalised NE proteins, and dysfunction of ER-shaping proteins are associated with many diseases including cancer and neurodegenerative disorders (Jevtic and Levy, 2014; Öztürk et al, 2020; Rose et al, 2022). Few physiological processes have been linked to ER–NE junctions, perhaps due to the lack of studies focusing on this particular type of junction. Our finding that ER–NE junctions have a distinct hourglass morphology in mammalian cells provides a basis for future molecular and functional studies in the context of physiology and disease.

## Methods

### Cell culture

Wild-type HeLa Kyoto cells (RRID: CVCL_1922) were incubated in Dulbecco's modified Eagle media (DMEM with low glucose, Sigma, D6046-500ML) supplemented with 10% foetal calf serum (FCS, Sigma, F7524-500ML) and 1% Penicillin-Streptomycin (Pen/Strep, Sigma, P4333-100ML). Cells tested negative for mycoplasma infection using a mycoplasma detection kit (Biological Industries, 20-700-20).

### Cell seeding and cell cycle synchronization for correlative light and electron microscopy

Sapphire discs (D6mm, Leica Microsystems) were washed for 2 h in HCl, rinsed once with tap water, and then rinsed several times with distilled water. After letting them dry, the discs were coated with a pattern-branded 10 nm layer of carbon (Edwards Auto306 high vacuum evaporator) and baked overnight at 120 °C in an oven. Carbon-coated sapphire discs were assembled on 1-well cups for SampLink chambers (Leica Microsystems). The assembled SampLink chambers were washed in 70% ethanol, exposed to UV light for 20–30 min, and dried overnight. Before seeding the HeLa cells, the discs were coated with 0.1 mg/ml poly-L-lysine hydrobromide

multimeric complexes and protein aggregates between the lumens of the ER and NE. Interestingly, Apomucin, which forms oligomers of 500–1000 kDa (Eckhardt et al, 1987), localises predominantly in the ER lumen rather than the NE lumen in mammalian cells (Deschuyteneer et al, 1988). ER–NE junctions may act as a size-selective gate that prevents large protein complexes and misfolded proteins to enter the perinuclear space (Sun and Brodsky, 2019). In this way, the constricted junctions may protect the NE from the potential toxicity of misfolded/aggregated proteins. The two types of ER–NE junctions we observed—with and without a luminal connection—raise the possibility that ER–NE junctions can change configuration. The membrane contact sites between the ER and other

(Sigma, P1274) for 1–1.5 h at 37 °C. Few hours after seeding the cells, when most cells had adhered to the sapphire disc, the media was replaced with 2 mM thymidine (Sigma, T9250-1G) in DMEM + FCS+Pen/Strep media overnight. On the next day, samples were washed twice in Dulbecco's phosphate-buffered saline (DPBS, Sigma, D8537) and incubated in fresh DMEM + FCS +Pen/Strep media to release cells from the first thymidine block. Six hours later, samples were washed in PBS and incubated in 2 mM thymidine in DMEM + FCS+Pen/Strep overnight. On the following day, samples were washed twice in PBS and incubated in fresh DMEM + FCS+Pen/Strep media for approximately 8 h to release cells from the second thymidine block. This double thymidine block does not affect the morphology of the ER and NE, judging from the following two data: (i) the ER junction width is comparable in double-thymidine treated and untreated HeLa cells in Figs. 2I and 5E (p-value > 0.1, unpaired two-tailed t-test), and (ii) the NE thickness has been reported to be similar between double-thymidine treated and untreated HeLa cells (Otsuka et al, 2016; Otsuka et al, 2018).

### Live cell imaging and high-pressure freezing

Approximately eight hours after the release from the second thymidine block, the media was replaced with imaging media (DMEM without Riboflavin and Phenol Red, Thermo Fisher Scientific, Gibco 041-96205M, containing 10% FBS, 1% Pen/Strep, and 50 nM SiR-DNA, Spirochrome, SC007). 1.5 h later, samples were transported to the light microscope using the Baker Ruskinn OxyGenie transportable incubator (Leica Microsystems) at 37 °C equipped with gas cylinders (Catalina Cylinders) filled with 5% $CO_2$ in a synthetic atmosphere of 20% $O_2$ and 80% $N_2$. Cells in the SampLink chambers were monitored in the light microscope THUNDER Imager Nano (Leica Microsystems) coupled with an incubator at 37 °C. During imaging, the SampLink chambers remained connected to the gas supply from the transportable incubator (Bragulat-Teixidor and Otsuka, 2024). Time-lapses of regions enriched in metaphase cells were acquired every 20 s in the bright-field and far-red (SiR-DNA) channels using a 20× 0.40 NA DRY NPLAN Epi objective (Leica Microsystems). When most dividing cells in the field of view reached late anaphase and telophase, samples were high-pressure frozen using the High Pressure Freezer Leica EM ICE equipped with the Coral Life workflow (Leica Microsystems). It took 1.0-1.5 min from the last time-lapse imaging until the high-pressure freezing, and the time lag was recorded to precisely determine the duration after anaphase onset. Cells were frozen in DMEM + FCS+Pen/Strep supplemented with 20% FICOL-PM400 (Sigma, F4375-25G) as a cryoprotectant, which was added to the samples few seconds before freezing. The lid carriers (Leica Microsystems) had a cavity depth of ~25 μm, and they had been in contact with 1-hexadecene (Merck, 8.22061.0100). After high-pressure freezing, sapphire discs were stored in liquid nitrogen until the subsequent sample preparation steps.

### Freeze substitution and resin embedding

Freeze substitution was performed in a Leica EM AFS-2 freeze substitution unit (Leica Microsystems) as described previously (Bragulat-Teixidor et al, 2022). Briefly, the samples were substituted in 0.1% uranyl acetate (UA), 2% osmium tetroxide (Electron Microscopy Sciences, 19134) and 5% $H_2O$ in acetone

(PanReac AppliChem, ITW reagents, 141007.1211) following this temperature ramp: −90 °C to −80 °C for 10 h, −80 °C to −30 °C for 10 h, −30 °C for 4 h, −30 °C to 0 °C for 6 h, 0 °C to 20 °C for 4 h, 20 °C for 5–6 h. Afterwards, samples were washed three times in pure acetone for at least 10 min each, and subsequently infiltrated with Agar 100 Epoxy resin (Agar Scientific). The resin infiltration was done progressively at room temperature with increasing concentrations of resin in acetone (3:1 for 2–3 h, 1:1 for 2–3 h, and 1:3 overnight). Infiltration with pure resin was done at room temperature for at least 5 h. Resin was polymerized at 60 °C for 72 h.

### Sectioning, gold bead attachment and post-staining

Resin blocks were sectioned every 250 nm using a Diamond knife (Diatome) and an ultramicrotome Leica ultracut UCT (Leica Microsystems). Sections were collected on Cu/Pd slot grids (Agar Scientific, G2564PD) coated with a film of 1% formvar (Agar Scientific) in chloroform (Sigma Aldrich, 32211-1L-M). To allow better alignment of dual-axis tomography, 15-nm gold beads conjugated with Protein A (Cytodiagnostics, AC-15-05-10) were attached to both sides of the grids. To enhance membrane contrast, sections were post-stained at room temperature with 2% uranyl acetate in 70% methanol for 10 min and with 3% Reynold's lead citrate in $H_2O$ (Delta Microscopies) for 7 min.

### Electron tomography

To minimize the variation in shrinkage among different cells upon exposure to the electron beam, we pre-irradiated the entire cells with the electron beam until the total electron dose reached ~1000 e/Å² so that the shrinkage occurred homogeneously throughout the cell. We verified that the variation of the shrinkage between different samples is minimal by comparing the nuclear pore diameter between different cells in our EM tomograms (Fig. EV3).

As ER–NE junctions are too tiny to be identified reliably on 2D micrographs, we developed a two-step tomography-based screen consisting of 'quick' and 'high quality' tomograms (Bragulat-Teixidor and Otsuka, 2024). The first step aimed at minimizing the acquisition time of the tilt series while allowing sufficient image quality to resolve the ER from NE membranes. These tilt series were acquired over a ± 20° tilt range with an angular increment of 2° at a pixel size of 0.567 nm. These tomograms were taken with slight overlaps to cover the entire NE in one or more sections per cell (Fig. 1B). These tilt series allowed the reconstruction of 'quick' tomograms, which were screened manually for potential ER–NE junctions. In the second step of the screen, 'high quality' tomograms (typically dual axis, from tilt series with a ±50° tilt range, an angular increment of 1°, and a pixel size of 0.451 nm) were taken at the regions where the 'quick' tomograms revealed potential ER–NE junctions. The reason why we took dual-axis tomograms is to minimize the missing wedge effect (Arslan et al, 2006) and to improve the image quality to better visualise the lipid bilayer.

We used a similar two-step tomography-based screen to search for ER–ER junctions with a similar topology to ER–NE junctions (i.e. an ER piece contacting to the flat side of an ER sheet) in the same cells where ER–NE junctions were analysed. We acquired 'low magnification' tomograms (from tilt series with a ± 50° tilt range,

an angular increment of 1°, and a pixel size of 1.137 and 2.181 nm) and 'high quality' tomograms (typically dual axis, from tilt series with a ± 50° tilt range, an angular increment of 1°, and a pixel size of 0.451 nm). The 'low magnification' tomograms allowed to find ER–ER junctions in the cytoplasm; the 'high quality' tomograms, to image the junctions at a high magnification for a better visualization and quantification.

The tomograms were reconstructed from tilt series with the R-weighted backprojection method implemented in the IMOD software package (version 4.11.7) (Kremer et al, 1996). The tilt series were acquired using the Serial EM software (v3.x) under the transmission electron microscope (TEM) Tecnai G2 20 operated at 200 kV and equipped with an Eagle 4k HS CCD camera. The Stitching plugin (Preibisch et al, 2009) in FIJI (Schindelin et al, 2012) was used to efficiently relocate the tomograms in stitched overviews of the cells. The contrast of EM images was enhanced by projections of 10 tomographic slices (corresponding to 4.51 nm).

## Quantification of the frequencies of ER–NE junctions

The frequency of ER–NE junctions was calculated by dividing the total junction number by the total NE surface area screened for junctions in quick tomograms. The junctions were identified in high-quality tomograms as described previously. The NE area was measured from quick EM tomograms, taking into account the overlay between consecutive overlapping tomograms. For telophase, we calculated the NE from meshes obtained by manually tracing the NE using IMOD. In general, we traced the NE on three tomographic planes (at the beginning, middle and end of the tomogram) and interpolated the NE traces along the tomographic axis. Since the NE is highly curving in telophase cells (Fig. 4A), the NE interpolation sometimes mismatched the raw EM data. Whenever there was a mismatch, additional contours were drawn on the NE so that the mesh could represent the true NE. For interphase, the NE is flat and almost perpendicular to the viewing plane of the section (Fig. 4B), so the NE area was estimated by multiplying the length of the NE in 2D overview images with the depth of the NE from the start to the end of the section. Concretely, we first identified regions with different tilts of the NE on 2D overviews of cells, and measured their length. Then, we measured the thickness of the NE in a quick tomogram of that region by a 2-point line in IMOD. Finally, the NE area was calculated by multiplying the length of a certain NE region with the depth of the NE in that same region. We confirmed the robustness of the method in three cells by comparing the NE surface area obtained in IMOD meshes (as done for telophase cells) versus the area obtained after multiplying the length with the depth of the NE on sections. The NE area values obtained by either method differed only up to ~3%, which confirms the reliability of this measurement.

## Measurement of the NE width below ER–NE junctions

The NE width was measured as a 2-point line spanning the perinuclear space below ER–NE junctions and at regions 200–500 nm away from ER–NE junctions and nuclear pore complexes (Fig. EV1D). The measurements were done in IMOD after orienting the tomograms in a way such that the NE would be horizontal and perpendicular to the viewing plane. Then, below junctions, the width was measured from the base of the junction

(where the ONM would be present if the junction would be absent) to the closest point at the inner nuclear membrane (INM). Away from junctions, the width was measured from a point at the ONM to the nearest point at the INM. The distance away from junctions was measured twice for each junction (i.e. in two opposite directions away from the junction).

## Measurement of the distance from ER–ER junctions to the NE

The distance from the neck of ER–ER junctions to the closest point at the outer nuclear membrane of the NE was measured as a 2-point line in IMOD. The measurements were done in 'low magnification' tomograms that contained both the ER–ER junction and the NE.

## Electron microscopy of yeast cells

Wild-type strains of budding yeast (*Saccharomyces cerevisiae*, BY4741, Euroscarf) cells were cultured at 30 °C in synthetic dextrose complete media. Asynchronous yeast cells were high-pressure frozen at the log phase and freeze substituted as described previously (Romanauska and Köhler, 2021). Sections with a thickness of 400 nm were cut using a Leica UCT ultramicrotome (Leica Microsystems). The grids were coated with 15-nm gold beads conjugated with Protein A on both sides, and the sections were post-stained with 2% UA in methanol and 3% Reynold's lead citrate in $H_2O$ as described above. Cells were observed using the Serial EM software (v3.x) under the transmission electron microscope (TEM) Tecnai G2 20 operated at 200 kV and equipped with an Eagle 4k HS CCD camera. At the regions where ER was visible near the NE, dual-axis tomograms were acquired typically over a −60° to +60° tilt range with an angular increment of 1° at a pixel size of 0.451 nm.

## Mesh generation

Meshes of the junctions were created from manual segmentations using the IMOD software. For the segmentation of ER–NE junctions in interphase and in telophase, we drew contours across the middle of the lipid bilayer of the ER, ONM and INM every 2.25–4.51 nm tomographic slices. For the segmentation of ER–ER junctions in interphase and ER-ER/NE junctions in anaphase, the membranes were segmented every 9.02 nm tomographic slices. After the manual segmentations, an algorithm in the IMOD software connected points of the consecutive contours to create a 3D mesh whose closed surface is comprised of multiple small triangles connected by common edges and vertices, and with a minimal total surface area (Kremer et al, 1996). Eventually, the meshes represent a 3D model of the manually segmented structure.

## Generation of side profiles of ER–NE and ER–ER junctions in tomograms

The side profiles were drawn by tracing manually the ER membrane and the ONM across the middle of the lipid bilayer after rotating the EM tomogram to the orientation of the sagittal plane of the junction in the Slicer mode of IMOD (Kremer et al, 1996; version 4.11.7).

## Cross-section analysis for determining top profiles, width, aspect ratio, and length of ER–NE and ER–ER junctions in tomograms

For each mesh, cross-sections were generated from the junction base at fixed intervals (typically 0.4 to 2.0 nm which varied for each example) along a centreline that spanned the neck of the junction using the NeuroMorph Analysis and Visualization Toolkit (Jorstad et al, 2018) in Blender (version 2.83) (Hess, 2010). The cross-sections allowed to obtain the top profiles, and to analyse the width, aspect ratio, and length of junctions. The top profiles were drawn manually at the neck of each junction after orienting the EM tomograms in the Slicer mode of IMOD (Kremer et al, 1996; version 4.11.7). The orientation and position of top profiles corresponded to the one of the cross-section with the minimum surface area within 25 nm away from the base of the junction. To identify the cross-section with the minimal surface area, the area of all cross-sections was measured using a custom script based on Python 3.9.5 using the libraries Trimesh 3.10.8 (Dawson-Haggerty et al, 2019) and Shapely 1.8.5 (Gillies et al, 2022). The width of junctions was defined as the average of the longest and shortest lengths of the bounding rectangle around the top profiles. The aspect ratio of junctions was calculated from the minor and major lengths of the bounding box for the cross-section. The bounding box was determined to be the minimum area rectangle allowing rotations, so that it takes care of any tilts in the projections. The width and aspect ratio measurements were done using a custom-made script. The length L of a junction resulted from the sum of two shorter length measurements named $L_1$ and $L_2$, where $L_1$ was the distance from the minimum area cross-section to the closest cross-section towards the NE with a 1.2-fold larger area and $L_2$ was the distance to the cross-section with a 1.2-fold larger area towards the ER. The measurements were done using a custom-made script.

## Stereology-based approach to look for ER–NE junctions in FIB-SEM datasets

We developed a stereology-based approach to search for ER–NE junctions in FIB-SEM datasets. We downloaded the datasets of a mouse pancreatic islet cell treated with high glucose (Data ref: OpenOrganelle jrc_mus-pancreas-1, 2019; voxel size (nm) of $4.0 \times 4.0 \times 3.4$ in x, y, z), a HeLa cell (Data ref: OpenOrganelle jrc_hela-2, 2017; voxel size (nm) of $4.0 \times 4.0 \times 5.2$ in x, y, z), and a human macrophage (Data ref: OpenOrganelle jrc_macrophage-2, 2018; voxel size (nm) of $4.0 \times 4.0 \times 3.4$ in x, y, z) from OpenOrganelle (Heinrich et al, 2021; Xu et al, 2021). Segmentations of the entire nucleus were required to uniformly distribute regions to search for junctions on the nuclear surface. The segmentations of the nuclei of the HeLa and the macrophage cells were available on OpenOrganelle. For the mouse pancreatic islet cell, we segmented the nucleus automatically doing a pixel classification in Ilastik (Berg et al, 2019; version 1.3). The automated segmentations were refined using filters and other options for binary images in FIJI so that the surface of the automatically-segmented nuclei would coincide with the NE on the raw data. For each nuclei, 52–56 points were selected at the surface of the automatically-segmented nuclei using a custom-written script. The first point was selected randomly, and the rest of points were distributed uniformly at the NE in a radial way from the randomly-positioned first point. Around each point, spherical regions with a radium of 800–1000 nm were screened manually to identify potential ER–NE junctions. For the mouse pancreatic islet cell and human macrophage, we measured the length of ER–NE junctions in all the 52 and 56 regions that were selected. For the HeLa cell, we quantified the junction length in 25 out of 52 regions. To compare ER–NE with ER–ER junctions, three-way junctions were searched in the ER at random regions of cytoplasm.

## Quantification of the width of ER–NE and ER–ER junctions in FIB-SEM datasets and in budding yeast tomograms

The widths were quantified in IMOD. For constricted junctions, the width corresponded to the length of a 2-point line traced across the neck of the junction at the sagittal plane. For non-constricted junctions, the width was measured with a 2-point line across the middle of the triangulated base of the junction at the sagittal plane.

## Sample size determination and statistical analysis

For electron tomography, we analysed 3 cells in late anaphase, 2 cells in early telophase, and 9 cells in interphase from one experiment. We screened the NE and the cytoplasm by tomography until we found approximately 15 ER–NE and ER–ER junctions. The exact value of the analysed surface area and the number of junctions that we found are described in the main text. The statistical analysis of junction morphology was carried out after all the data were acquired. For the FIB-SEM analysis of three cells, ER–NE junctions were inspected in 52–56 uniformly-distributed regions of interest on the NE. We picked up all the potential ER–NE junctions and did not perform any data exclusion. For each cell, we found 23–34 ER–NE junctions, and we looked for a similar number of ER–ER junctions at the cytoplasm of the same cell. Statistical analyses were performed only after all the data were obtained. Statistical analysis methods, sample sizes and $p$-values for each experiment are indicated in figure legends. No randomization and blinding was done in the analysis.

# Data availability

Raw 2D-tilt series EM images for tomography and the reconstructed tomograms are deposited to Electron Microscopy Data Bank and, the Electron Microscopy Public Image Archive. The tomograms containing ER–NE junctions in interphase: https://www.ebi.ac.uk/emdb/EMD-50068, https://www.ebi.ac.uk/pdbe/emdb/empiar/12025; those of ER–ER junctions in interphase: https://www.ebi.ac.uk/emdb/EMD-50115, https://www.ebi.ac.uk/pdbe/emdb/empiar/12050; those of ER–NE junctions in telophase: https://www.ebi.ac.uk/emdb/EMD-50110, https://www.ebi.ac.uk/pdbe/emdb/empiar/12048; those of ER–ER/NE junctions in anaphase: https://www.ebi.ac.uk/emdb/EMD-50134, https://www.ebi.ac.uk/pdbe/emdb/empiar/12051. The code to quantify the junction ultrastructure is available at https://github.com/hbragulat/ER-NEjunction3Dmorphometry.

The source data of this paper are collected in the following database record: biostudies:S-SCDT-10_1038-S44319-024-00175-w.

## Peer review information

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

## Acknowledgements

This project was supported by laboratory startup funding from the Medical University of Vienna, by the Vienna Science and Technology Fund (WWTF; project LS19-001), and by the Austrian Science Fund (FWF; project P 36743-B) to SO. H.B.T. acknowledges a DOC Fellowship of the Austrian Academy of Sciences and a Max Perutz PhD Fellowship from the University of Vienna and the Medical University of Vienna. We thank the electron microscopy facilities at the Vienna BioCenter (Thomas Heuser, Marlene Brandstetter, Nicole Drexler, Sonja Jacob, and Harald Kotisch) and EMBL Heidelberg (especially Martin Schorb and Paolo Ronchi) for technical support and discussion. In addition, we thank Paul Wurzinger and Robert Kirmse (LEICA Microsystems) for technical support. Yeast samples were kindly provided by Anete Romanauska (Alwin Köhler's lab). We acknowledge the members of the labs of Shotaro Otsuka (especially Clara-Anna Wagner, Tamara Völkerer, Julia Scholz, Kaike Ren, Pallavi Deolal, Clemens Geppner, Pauline Vinet, Simay Tekin, and Nikoleta Kavaja), Alwin Köhler, Elif Karagöz, Roland Foisner, Peter Fuchs, David Haselbach, and Life Science Editors for feedback and discussions. Eija Jokitalo also provided us with valuable references.

## Author contributions

**Helena Bragulat-Teixidor**: Conceptualization; Data curation; Formal analysis; Funding acquisition; Investigation; Visualization; Methodology; Writing—original draft; Writing—review and editing. **Keisuke Ishihara**: Conceptualization; Data curation; Formal analysis; Investigation; Visualization; Methodology; Writing—original draft; Writing—review and editing. **Gréta Martina Szücs**: Data curation; Formal analysis; Visualization. **Shotaro Otsuka**: Conceptualization; Data curation; Formal analysis; Supervision; Funding acquisition; Investigation; Visualization; Methodology; Writing—original draft; Project administration; Writing—review and editing.

Source data underlying figure panels in this paper may have individual authorship assigned. Where available, figure panel/source data authorship is listed in the following database record: biostudies:S-SCDT-10_1038-S44319-024-00175-w.

## Disclosure and competing interests statement

The authors declare no competing interests.

# Expanded View Figures

**Figure EV1.  Additional analyses of ER and NE morphology around the junctions in interphase.**

(**A**) Additional examples of ER–NE junctions and different types of ER–NE contact sites in interphase. Electron tomographic slices at the sagittal plane of ER–NE junctions with and without lumen, ER–NE 'contact sites' where the ER membrane is juxtaposed to the ONM but their membranes are not fused, and where the ER is connected to the ONM via filamentous-like densities. For each junction, the left image shows raw EM data; the right one, the EM data on which the ER and the NE are coloured in orange and green, respectively. The filaments are indicated with magenta arrowheads. Scale bars: 20 nm. ER: endoplasmic reticulum, NE: nuclear envelope. See also Movie EV3. (**B**) Membrane profiles of the side views of ER–NE junctions with and without lumen found in interphase cells. To distinguish individual junctions, the profiles are coloured differently for each junction. (**C**) Morphology of the ER at 100 nm away from the junction base that connects to the NE and to the flat sheet of the ER that were analysed in this study. (i) The 3D meshes of the junctions in which the ER pieces connecting to the flat membrane are tubule-like, sheet-like, and with other morphology. Their 90°-rotated views are shown below. White arrowheads point to regions 100 nm away from the junction base. Note that in tubules the width stays similar after 90° rotation while in sheets the width gets much thinner. Scale bar: 100 nm. (ii) The proportion of each morphological type of the ER connecting to the NE (left) and the flat sheet of the ER (right). (**D**) Local NE dilation below ER–NE junctions. (i) A scheme depicting that the width of the perinuclear space was measured below ER–NE junctions and 200–500 nm away. (ii) Quantification of the NE width below ER–NE junctions (with and without lumen, $n = 10$ and 9, respectively) and 200–500 nm away from each junction ($n = 20$ and 18, respectively) from 9 cells from a single experiment. *$p$-value < 0.05; two-tailed Mann–Whitney test. n.s.: not significant ($p$-value > 0.5). The median is depicted as a horizontal line. (**E**) Distance from the ER–ER junctions that were analysed in Fig. 2F–I to the outer nuclear membrane (ONM). $n = 14$ junctions from 4 cells from a single experiment. The median is depicted as a horizontal line. (**F**) An ER piece forming simultaneously a constricted junction to the NE and a wide junction to an ER sheet. The left image shows raw EM data; the middle one, the EM data on which the ER and the NE are coloured in orange and green, respectively. The right image shows the 3D mesh of the ER and the NE. The ER–ER and ER–NE junctions are indicated by white arrowheads. Scale bar: 200 nm. C: cytoplasm, N: nucleus. See also Movie EV4.

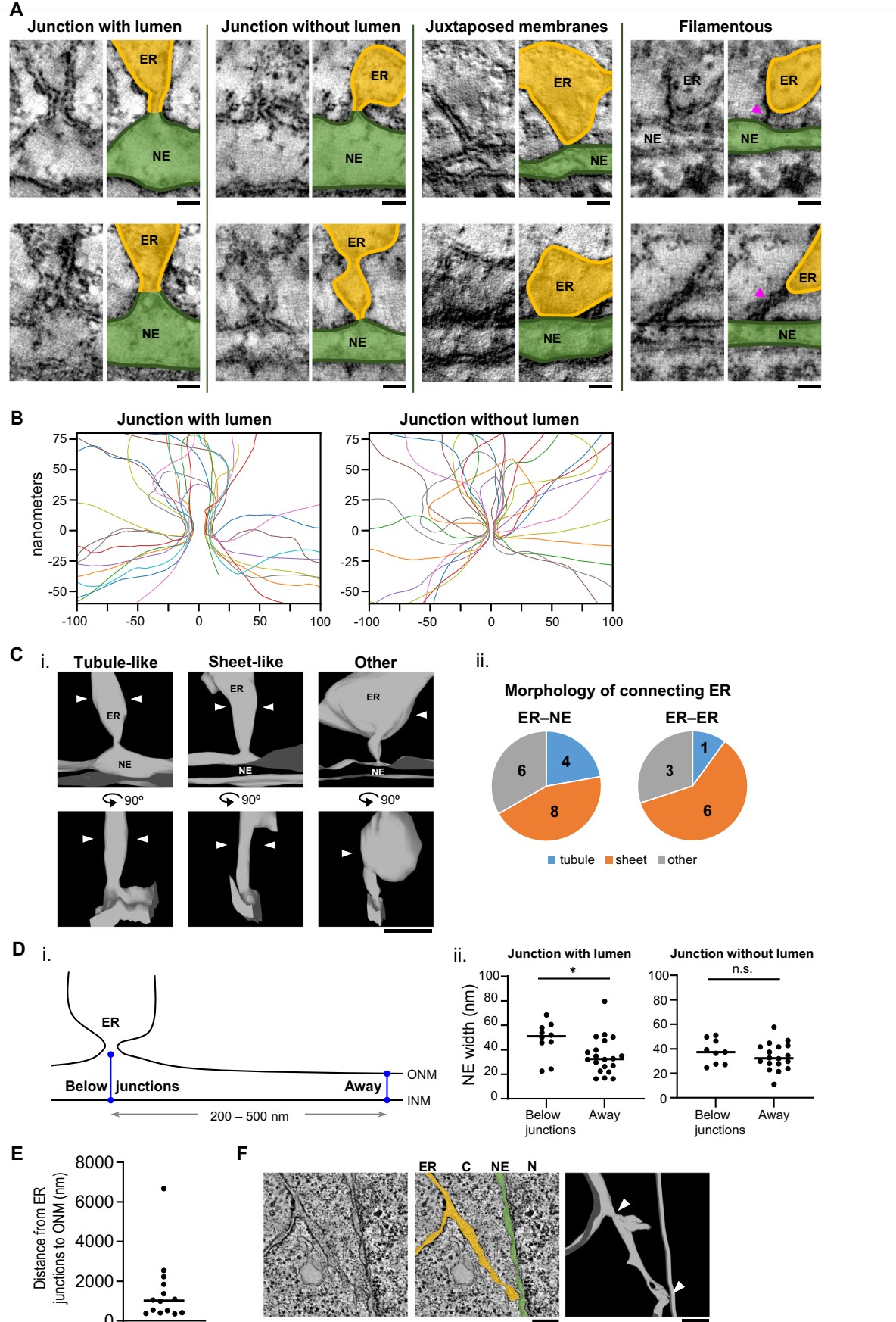

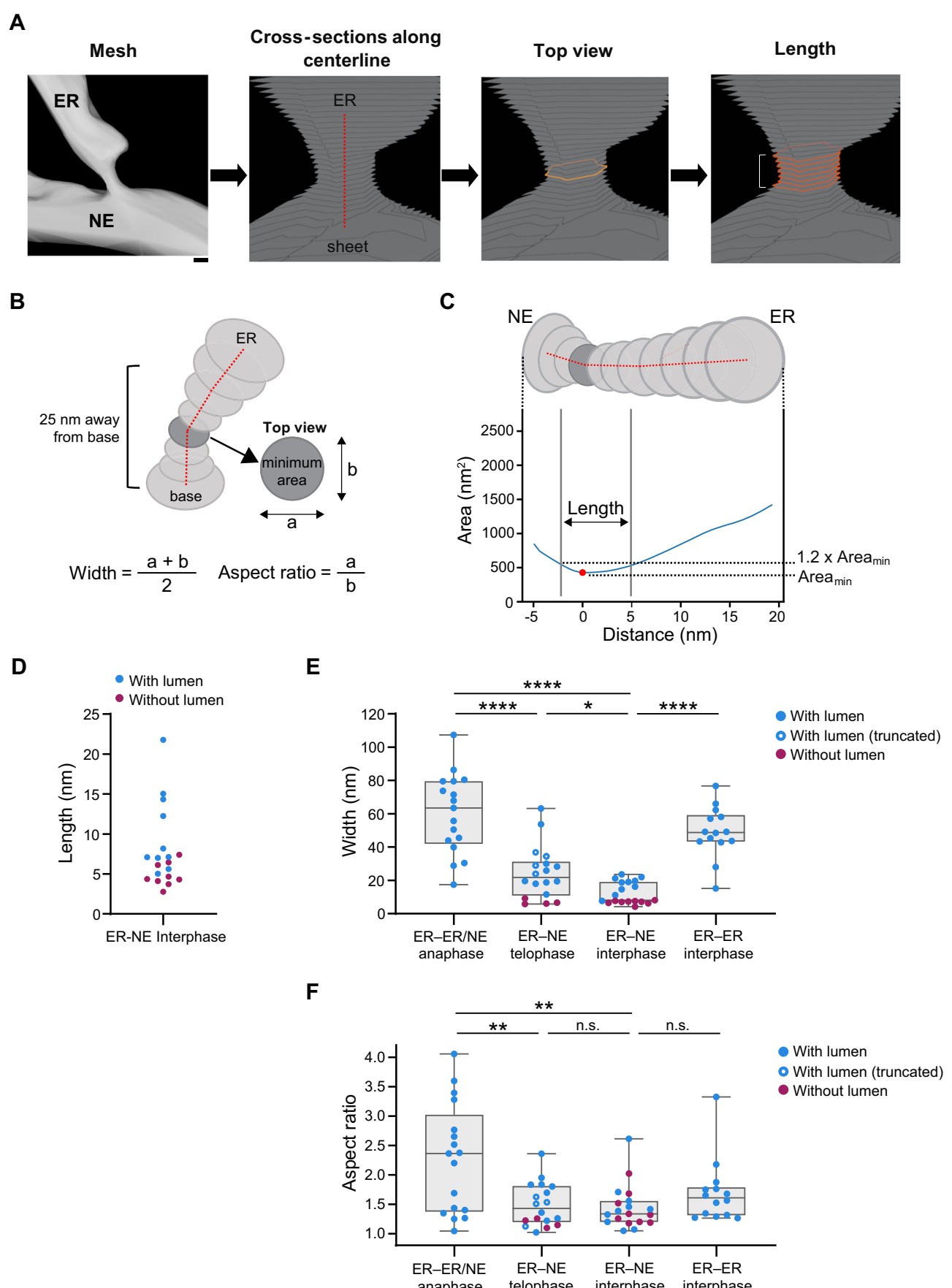

**Figure EV2. 3D-ultrastructural analysis of junctions.**

(A) To quantify the 3D ultrastructure of the junctions, cross-sections were obtained at regularly-spaced intervals along a centreline of the 3D meshes. The cross-sections allowed identifying the junction top view and length. Scale bar: 20 nm. (B) The cross-section with the minimum surface area was selected up to 25 nm from the base for making a top-view profile. The schemes represent how to determine the width and the aspect ratio of junctions. (C) A plot showing the cross-sectional area along the junction axis from the NE to the ER. The curve is centred at the cross-section with the minimum surface area. The distance between the two cross-sections with a surface area 1.2 times larger than the minimum one was defined as the length of the junctions. (D) Length of ER–NE junctions in interphase ($n = 19$ from 9 cells from a single experiment). The plots for the junctions with and without lumen are colour-coded in blue and magenta, respectively. (E) Combined plots of the width of ER–ER/NE junctions in anaphase, ER–NE junctions in telophase and interphase, as well as the width of ER–ER junctions in interphase. $n = 17$, 18, 19, and 14 for ER–NE anaphase, telophase, interphase, and ER–ER interphase, from 3, 2, 9, and 4 cells, respectively, from a single experiment. Dots with a white filling indicate the 4 truncated top view profiles, whose width is underestimated. *$p$-value < 0.05, ****$p$-value < 0.0001; two-tailed Mann–Whitney test. Centre line, median; box limits, upper and lower quartiles; whiskers, min and max. (F) Combined plots of the aspect ratio of the junctions. The plots are shown as in (E). $n = 17$, 18, 19 and 14 for ER–NE anaphase, telophase, interphase, and ER–ER interphase, from 3, 2, 9, and 4 cells, respectively, from a single experiment. **$p$-value < 0.01; two-tailed Mann–Whitney test. n.s.: not significant ($p$-value > 0.5).

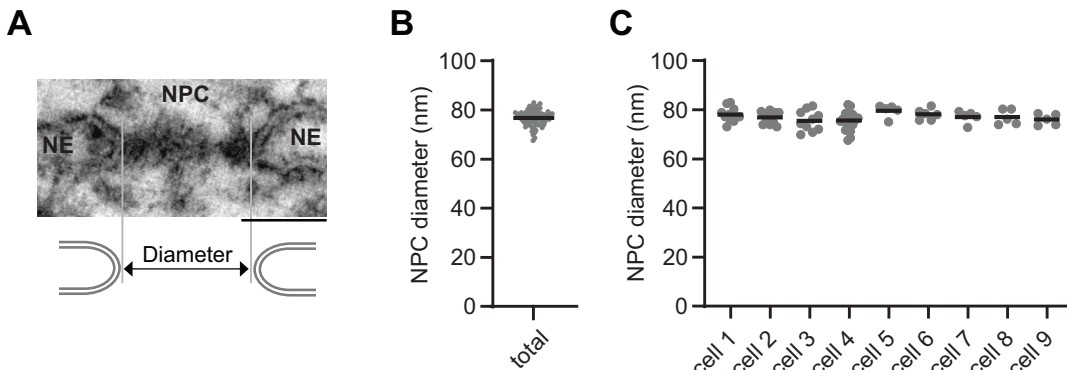

**Figure EV3. Quantification of nuclear pore diameter to assess the degree of shrinkage in the EM tomograms.**

(A) An electron tomographic slice of a nuclear pore in interphase. Average intensity projection of 20 z-slices (corresponding to 9 nm). Scale bar: 50 nm. Inner nuclear pore diameter was measured as indicated by a bidirectional arrow in the bottom panel. NPC: nuclear pore complex, NE: nuclear envelope. (B) Nuclear pore diameter in the EM tomograms in which we inspected ER–NE and ER–ER junctions. The average diameter was 77 nm ± 3.2 nm (mean ± S.D., 80 pores in 9 cells from a single experiment). Since the nuclear pore diameter was 92 nm in cryo-EM tomograms of interphase HeLa cells that were vitrified by plunge freezing (Mosalaganti et al, 2022), the shrinkage of the specimen in our EM tomograms is estimated to be 17%. (C) Comparison of nuclear pore diameter among the interphase cells analysed by EM tomography in this study. The diameter was comparable between different cells ($n = 12, 12, 10, 20, 5, 6, 5, 5, 5$ nuclear pores in 9 cells from a single experiment), indicating that the shrinkage occurred to a similar degree in the EM tomograms of cells that we inspected.

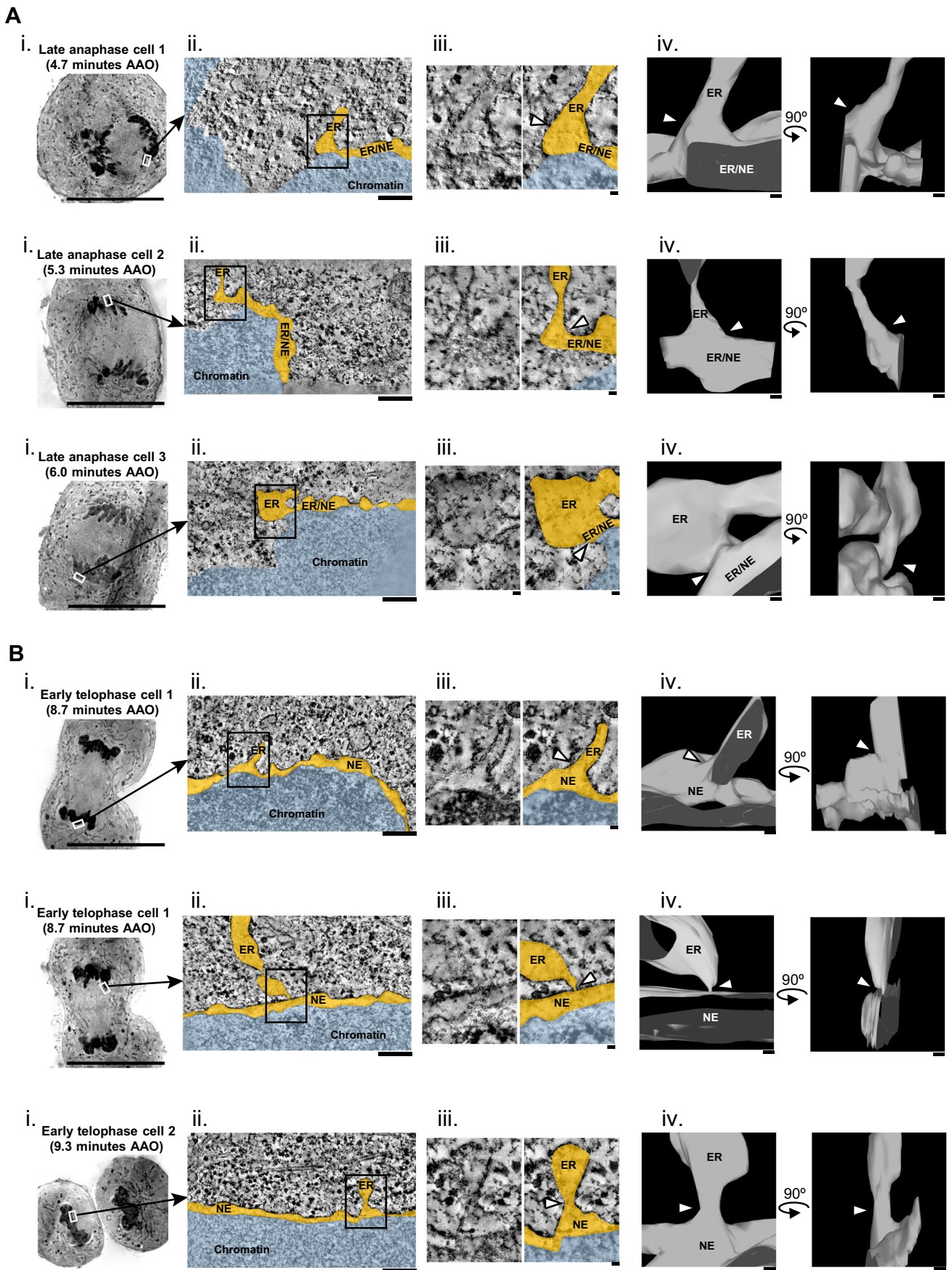

**Figure EV4. Additional examples of junctions in late anaphase and in early telophase.**

(**A**) Additional example images of ER junctions contacting the chromatin in late anaphase cells. (i) 2D EM micrographs of three different late anaphase cells. AAO: After Anaphase Onset. (ii) Tomographic slices of cells in late anaphase showing that the ER (orange) starts to contact the chromatin (blue), but the NE is still indistinguishable. (iii) Enlargement of the regions indicated in (ii). For each junction, the left image shows raw EM data; the right one, the EM data on which the ER and the chromatin are coloured in orange and blue, respectively. (iv) 3D meshes of the junctions and their 90°-rotated views. Junctions are indicated by white arrowheads in (iii) and (iv). (**B**) Additional example images of ER–NE junctions in early telophase. Images are displayed in the same way as in (**A**). Data Information: Scale bars for (i): 20 μm; Scale bars for (ii): 200 nm; Scale bars for (iii, iv): 20 nm.

1. Whole cell FIB-SEM dataset from public database

2. Automated segmentation of **NE**
3. Uniformly distributed **regions of interest**

4. Look for ER-NE junctions

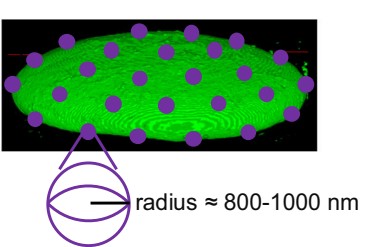

radius ≈ 800-1000 nm

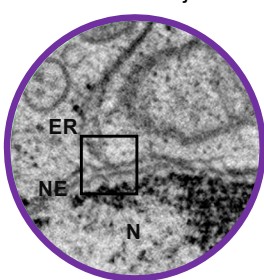

**Figure EV5. Stereology-based approach to find ER–NE junctions in FIB-SEM datasets of entire cells.**

The datasets were downloaded from OpenOrganelle (Heinrich et al, 2021: Data ref: Heinrich et al, 2021; Xu et al, 2021: Data ref: Xu et al, 2021). On the surface of automatically-segmented nuclei, the regions of interest were uniformly sampled with a radius of 800–1000 nm. In these regions, potential ER–NE junctions were searched manually.

