## [Peer Review File · EMBO Reports]

The endoplasmic reticulum connects to the nucleus by constricted junctions that mature after mitosis

Helena Bragulat-Teixidor, Keisuke Ishihara, Greta Szucs, and Shotaro Otsuka

Corresponding author(s): Shotaro Otsuka (shotaro.otsuka@univie.ac.at) , Helena Bragulat-Teixidor (helena.bragulatteixidor@meduniwien.ac.at)

Review Timeline:

Transfer Date:	11th Dec 23
Editorial Decision:	15th Dec 23
Revision Received:	8th Apr 24
Editorial Decision:	15th May 24
Revision Received:	27th May 24
Accepted:	3rd Jun 24

Transaction Report: This manuscript was transferred to EMBO reports following peer review at Review Commons.

**Review
COMMONS**

Review #1

1. Evidence, reproducibility and clarity:

Evidence, reproducibility and clarity (Required)

The mechanisms that differentiate ER from the nuclear envelope (NE) remain to be fully elucidated but likely depend at least in part on junctions between the ER and NE. How such junctions are formed and maintained is the subject of this manuscript where extensive correlative light and electron microscopy is used to observe and characterize ER-nuclear envelope (ER-NE) junctions at distinct phases of the cell cycle. The authors make use of their own electron tomography data as well as publicly available focused-ion beam scanning electron microscopy (FIB-SEM) datasets to compare the morphology of these junctions in different human cell types as well as in budding yeast. The major finding is that ER-NE junctions in human cell lines are more constricted than ER-ER junctions, often to the point of excluding lumen. The examination of mitotic cells suggests that this constriction likely occurs at the end of mitosis as the NE is completing its maturation from ER to NE. The implications of these morphological changes are discussed but there are no mechanistic or functional studies. Overall, the data are well presented, are of high quality and are rigorously evaluated. The manuscript is well written and scholarly, and the speculations as to the function of the constrictions are reasonable. I only have minor comments.

1. In Figure 2D, the authors present evidence to demonstrate that an hourglass-like constriction occurs at ER-NE junctions. From the side view, it is difficult to interpret this on the plot, particularly for the ER-NE junctions with a lumen. Perhaps, in the supplemental data, the authors could plot both with and without lumen data separately, and color-code individual traces? I believe this would convey the hourglass nature of these constrictions more clearly.
2. In the Methods section, the authors should describe how carbon-coating of sapphire discs was achieved. If these were provided from the manufacturer precoated, this should be specified.
3. On page 10, Figure 5F callout 9 lines from the bottom likely should be 5E.

2. Significance:

Significance (Required)

Overall, this work provides an important new morphological perspective on the nature of ER-NE junctions in human cells. As the authors describe in their introduction, such

junctions have been noted previously in the literature but not in a dedicated study using modern imaging techniques in human cell lines. In describing the morphology of these junctions, the authors lay the groundwork for future mechanistic, functional, and structural studies.

3. How much time do you estimate the authors will need to complete the suggested revisions:

Estimated time to Complete Revisions (Required)

(Decision Recommendation)

Less than 1 month

No

Review #2

1. Evidence, reproducibility and clarity:

Evidence, reproducibility and clarity (Required)

****Summary:****

In this manuscript, Bragulat-Teixidor et al., use correlative live-cell imaging and electron tomography to study the structure of the endoplasmic reticulum-nuclear envelope (ER-NE) junction in HeLa cells (and also in *S. cerevisiae*). The authors also

make use of publicly available whole-cell FIB-SEM datasets to study ER-NE junctions in mouse pancreatic islet, HeLa, and human macrophage cells to corroborate their findings in other cell types.

The authors show that the structure of the ER-NE junction in interphase cells adopts an hourglass shape with a constricted neck. Comparing the ER-NE junction to the ER tubule-sheet junction, the authors show that these structures are different: the ER tubule-sheet junction is not constricted. Because the NE forms from the ER during postmitotic NE assembly, the authors compare the structure of the ER-NE junctions in anaphase, telophase, and interphase cells, and find that the junction becomes constricted in telophase. The number of ER-NE junctions increase going from telophase to interphase.

While the authors do not provide any direct evidence for this, they propose a functional model where the ER-NE junction is constricted because it regulates the supply of certain lipids and proteins from the ER to the NE. One proposed example is that the constriction of the ER-NE junction might prevent the passage of large protein aggregates from entering the NE.

The general question of how the structure of the ER-NE junction might regulate the passage of lipids and proteins from the ER to the NE is interesting and potentially important. However, the authors should address the following issues to improve the accuracy and completeness of this manuscript for it to be considered for publication.

****Major comments:****

1. The authors compare the structure of the ER-NE junction to the structure of the ER tubule-sheet junction in interphase cells. They should instead or in addition be comparing the ER-NE junction to ER sheet-sheet junctions. This is likely a better comparison for two reasons:

i) The NE is similar to an ER sheet due to its flat and extended structure. The ER membranes surrounding the NE consists mostly of a dense network of sheet-like ER (Zheng et al., 2022, PMID: 34912111). Therefore, the ER-NE junction should be compared to these NE-adjacent ER sheet-sheet junctions and not ER tubule-sheet junctions which are likely to be found in the cell periphery.

ii) In HeLa cells, the NE assembles from large ER sheets and not ER tubules (Zhao et al., 2023, PMID: 37098350; Otsuka et al., 2018, PMID: 29323269; Lu et al., 2011, PMID: 21825076). Therefore, the ER-ER junctions the authors are already studying in anaphase cells are likely to be ER sheet-sheet junctions, which should be kept the

same in their analysis of the ER-ER junctions in interphase cells.

Related to this point, comparing the side view panels in Figure 2D with 2H, it seems that the width of the ER membranes on either side of the neck region of the ER-NE junction is in fact getting wider (more sheet-like). This is in contrast to the ER-ER junction where the width stays constant for the ER tubule that is fusing onto the ER sheet. This suggests that indeed, the ER-NE junction is more similar to an ER sheet-sheet junction.

2. The authors claim that in late anaphase cells, the ER-ER/NE (written like this because the ER and NE cannot be distinguished like the authors also point out) junctions are not constricted and had a similar morphology to ER-ER junctions in interphase. However, this claim is only qualitative at the moment, as the authors do not provide any quantification of the width of the ER-ER/NE junctions in late anaphase cells. To make the current claim that the ER-NE junction only becomes constricted in telophase, the authors should report the width of the ER-ER/NE junctions in late anaphase cells.

In late anaphase cells, large ER sheets initially wrap around chromatin at the periphery of the chromosome mass (Zhao et al., 2023, PMID: 37098350; Otsuka et al., 2018, PMID: 29323269; Lu et al., 2011, PMID: 21825076). Therefore, the authors might find it easier to identify ER-ER/NE junctions in the so-called "non-core" regions, instead of in the current regions shown in Figure 3A.

****Minor comments:****

1. In the Supplementary Figures 1 A-D, make the scale bars white. Currently, the black scale bars are especially difficult to see in the top panels in Supplementary Figure 1C.
2. In the Results section entitled "The number of ER-NE junctions per cell increases from telophase to interphase", the authors should tone down this claim because the number of telophase cells examined is low (only 2 telophase versus 9 interphase cells). It would be better to include the word "slightly" in the title to change it to "slightly increases".
3. In the Results section entitled "The number of ER-NE junctions per cell increases from telophase to interphase", the authors state "These densities were much lower than those of ER-ER junctions...". For sure this is true for ER tubule-tubule junctions in the periphery of the cell as ER tubules form an intricate network by constantly fusing to each other, but it's not clear if this is also the case for ER tubule-sheet or ER sheet-sheet junctions. For clarity, the authors should state that they mean ER tubule-tubule junctions.

Same comment also for the statement "...although their abundance remains considerably lower than that of ER-ER junctions or nuclear pores at both cell cycle stages". The authors should state that they mean ER tubule-tubule junctions.

4. In the Results section entitled "The constricted morphology of ER-NE junctions is observed in different mammalian cells, but not in budding yeast", the authors state "...pancreatic islet cells (Figure 5A), HeLa (Figure 5B), and macrophage (Figure 5C) were significantly smaller than most ER-ER junctions (Figure 5F)". The last figure reference here is wrong and should be changed to Figures 5D-E.

5. In Discussion, the authors state "Proteins known to form and stabilize junctions in the ER, including Atlastins and Lunapark...". The authors should specify that they mean ER tubule-tubule three-way junctions. Also more generally throughout the manuscript, the authors should be more careful in specifying which ER-ER junctions they mean in each case.

6. In Discussion, the authors state "Thus, we favour a second scenario in which ER-NE junctions are generated from ER tubules that contact and eventually fuse with the ONM". Given that the ER membranes adjacent to the NE are mostly sheet-like (as pointed out in Major comment 1 above), the authors need to explain how they think an ER tubule (mostly found in the cell periphery) could access and fuse to the NE.

2. Significance:

Significance (Required)

Although the ER-NE junction has been studied in other organisms before, this study represents the first structural characterisation of the ER-NE junction in mammalian cells. Therefore, this study represents an advance for the field in gaining a better understanding of different ER structures and morphologies. How the ER is remodelled during the cell cycle is also an interesting question and an active field of research (Merta et al., 2021 PMID: 34853314; Zhao et al., 2023, PMID: 37098350) which this study further contributes to. This study would therefore be interesting for anyone interested in ER structure/morphology, ER-NE connections, and cell cycle regulation of such ER-NE connections.

My field expertise is in ER and NE. I do not have sufficient expertise to evaluate the methodology for the EM tomography part of this paper.

3. How much time do you estimate the authors will need to complete the suggested revisions:

Estimated time to Complete Revisions (Required)

(Decision Recommendation)

Between 3 and 6 months

No

Review #3

1. Evidence, reproducibility and clarity:

Evidence, reproducibility and clarity (Required)

The manuscript by Bragulat-Teixidor et al. is a study of the connection of the ER with the nuclear envelope. It uses advanced ultrastructural techniques: high pressure freezing instead of chemical fixation and EM tomography instead of serial sectioning. Synchronized HeLa cell cultures were examined during interphase, late anaphase (4-6 min after anaphase onset) and early telophase (8-10 minutes after anaphase onset).

The investigators find an unexpected, unusual structure - a constricted neck 7-20 wide and about 10 nm long where the ER connects to the nuclear envelope. The 7 nm connections had no apparent lumen. These are not seen in late anaphase when the NE has not yet formed, but they are seen a few minutes later during early telophase when there is a newly formed NE surrounding the chromosomes. A quantitation was made of their abundance, more was found later during interphase, and with wider lumens.

It is very nice to show the EM images as uncolored and segmented (colored). The images shown in the figures are presumably the best that were obtained during the

study. Heavy metals do not stain membranes uniformly or exclusively, and identification of structures doesn't always seem unambiguous. The three dimensional information can certainly make this easier though this information is difficult or not possible to show in journal format. In the end, the reader must depend on the judgment of the person who did the analysis. Overall, the analysis seems trustworthy.

HeLa cells are very convenient for getting information on cell cycle dependence. However, they are cancer cells in culture, so it is important to look at other cell types as well. The same methodology was used on budding yeast and they saw a wide tentlike connection, which reproduces an earlier study. This seems more consistent with what is known or expected from ER membranes. It is not less interesting but perhaps less puzzling.

To get evidence on other mammalian cells, the authors did an analysis of data from OpenOrganelle. These are high pressure frozen cells / tissue imaged by FIB-SEM. The voxels are 4 nm, which is significantly larger than those in EM tomography. Unfortunately, the difficulty of identifying structures is correspondingly more significant. The images shown do not contradict the HeLa results but by themselves (without the HeLa cell data), a convincing case for narrow connections probably couldn't be made.

The work in this manuscript seems to have been done well. Assuming that this structure is confirmed in other mammalian cells, another kind of question comes to mind: is this the final word on ER to NE connections? The lumenless neck does not seem like it would be a stable structure, somehow it seems like a transient one. In the future, it would help if a new structural protein was identified or some theoretical analysis to help explain the shape.

It is generally now assumed that high pressure freezing preserves structure perfectly. However, in this reviewer's mind, there is a possibility that some structures are not. The sample is brought to 2000 atmospheres within a few milliseconds, frozen, then the high pressure is released after a second. Although many intracellular structures do seem well preserved, could the junction be susceptible to high pressure? A second source of uncertainty is that in order to embed the samples in resin, the water was removed by freeze substitution. This is known to cause a small amount of tissue shrinkage and possibly could alter a delicate structure. Another way to look at this kind of structure is cryo-EM tomography on hydrated lamellae from plunge frozen cells. I don't recommend that the authors do another arduous, possibly too arduous set of experiments with a completely different technique, but perhaps another group has data which could support their findings.

The following are suggestions for the Discussion:

Yeast have many of the same biochemical processes as mammalian cells. Perhaps their lack of narrow connections can be used as a clue to the function of the narrow necks seen in HeLa cells. For instance, the authors speculate that the narrow connection serves to keep phosphatidylserine in the nuclear envelope low. If the yeast nucleus has the same concentration of phosphatidylserine as the ER, it would provide good evidence for this idea.

There might be other instances of lumenless neck structures. Dynamin mutants can cause a stable constricted tubule - are the dimensions of this tubule similar to that of the ER / NE connections? Or possibly some ESCRT related structure?

There do not seem to be any recent studies of the ER / nuclear membrane connection in fixed cells. However, there is serial section data online which can be inspected. There are connections in mouse brain cortex in the data of Kasthuri et al., 2015 (<https://neurodata.io/project/ocp/>). Instead of a tubule connection, there seems to be a narrow sheet of ER that connects to the nuclear envelope. But there is something odd about these too. The authors may like to mention something about this or similar work in their manuscript. This reviewer has looked at chemically fixed data from several cell types from his own unpublished data and connections are surprisingly hard to find. Possibly, the connection is particularly sensitive to chemical fixation.

2. Significance:

Significance (Required)

This is a careful and thorough study of the connection between the ER and the nuclear envelope. The discovery of reticulons and similar proteins, along with biophysical modeling, made the form of the ER accessible to analysis. The factors that govern ER structure are now much better understood. This is particularly true of sheets versus tubules, the three way tubule junctions and to some extent, the junction of ER tubules coming out of the edge of a sheet. However, with all this activity, the subject of the connection of the ER to the nucleus has not been examined in detail. What makes it different is that the tubule is connected perpendicular to the plane of a sheet.

The manuscript uses the best ultrastructural techniques and provides strong evidence for a narrow neck at this connection in HeLa cells. With the same methodology, yeast cells (*S. cerevisiae*) have a wider connection. OpenOrganelle data from other mammalian cell types was examined. This data has less resolution and although it does not contradict the HeLa cell data, it does not support it strongly.

This work is of interest to cell biologists specializing in membranous organelles or those interested in nuclear physiology. The connection of ER to nuclear envelope is an interesting problem that has not been studied recently. This manuscript could very well serve as a starting point for further structural or functional work by the authors or other groups.

3. How much time do you estimate the authors will need to complete the suggested revisions:

Estimated time to Complete Revisions (Required)

(Decision Recommendation)

Less than 1 month

Yes

Review #4

1. Evidence, reproducibility and clarity:

Evidence, reproducibility and clarity (Required)

****Summary:****

Membrane bound ribosomes and ER exit sites are present in the cytosolic side of

nuclear envelope (NE), suggesting that NE shares protein translocation, folding and quality control functions with the endoplasmic reticulum (ER). Moreover, membrane continuity between the ER and outer NE membrane is evident, and, thus, NE is considered as a subdomain of the ER. To support this, during cell division, NE loses its identity, and participates to daughter cells as part of the ER. However, NE has also membrane proteins and luminal proteins that are enriched to NE and absent from ER during interface, and the segregation of NE specific proteins/lipids occurs concomitantly with NE formation during late anaphase/telophase. In this study, the ultrastructure of the ER-NE junctions is described using high resolution electron tomography. Results show convincingly a specific constriction at the ER-NE neck during interface in several mammalian cell types. This structure is absent during metaphase, and also from the budding yeast. Authors present a model for the formation of ER-NE junctions in higher eukaryotes and speculate about their functional role.

****Major comments:****

The main conclusion of the paper is that although the ER and outer NE membranes are continuous, a specific hourglass shaped constriction at the neck is found in higher mammalian cells during interphase. The structure is specific to ER-NE necks, as it is absent during metaphase and ER-ER junctions. For the analysis, authors used high pressure freezing to ensure best structural preservation. Unfortunately, fixation is not the only potential source of artifacts; during tomography at ambient temperature, the thinning of the plastic sections under the beam can be up to 30%. In evaluation of the results, authors should consider how this thinning could affect the measurements of membrane distances and luminal width, and what type of distortions may happen as a consequence of asymmetric shrinkage.

In addition to analysis of own samples, authors took advantage of the publicly available whole-cell datasets in OpenOrganelle and used these datasets to expand the number of cell types analyzed. Moreover, the 3D-datasets were generated with different imaging technique, FIB-SEM. Although this technique provides lower resolution in general, it provides isotropic resolution, and the data could be used to eliminate the shortcomings of the tomography, thinning of the sections and the missing wedge. The authors could expand the comparison of the data from these different sources from this perspective, especially since HeLa cells were used in their own tomography studies and FIB-SEM datasets in OpenOrganelle. Similarly, it would be interesting to see if similar approach could be used to compare their results to those obtained by cryo-EM by utilizing the cryo-EM database. Have authors checked if any suitable datasets for analysis of ER-NE junctions could be found from public archives?

For the analysis of mitotic cells, double thymidine block was used to synchronize the

cell culture. It is not clear, why synchronization was necessary, as CLEM was used to select the cells, and their number was rather low. Do cells continue growing and synthesizing new proteins during thymidine blocks? As one way to control potential artifacts due to the synchronization treatment, authors could compare the average thickness of ER and NE in naturally occurring interphase and mitotic cells vs. synchronized cells.

****Minor comments:****

On page 5, last chapter (+ Fig.1 legend and materials and methods): "the quick tomograms covered the entire NE" is misleading, as the imaging covered a thin layer of the entire NE only. - Authors could have analyzed the entire NE from the FIB-SEM datasets but chose to use stereological approach to minimize their work.

To save time from the readers to follow the reference, authors could describe how the specimens used in OpenOrganelle datasets were fixed and processed, especially as they emphasize the importance of high pressure freezing in their own sample prep. Similarly, in Fig.4 legend, authors refer to measurements done in the previous study without explaining how and from what type of data.

Is there a difference between mesh generation and segmentation, or is it just two different terms used for the same thing by different programs?

2. Significance:

Significance (Required)

General assessment:

ER-NE gates were described earlier in the literature for specific cell types using standard thin-section TEM imaging, and in this study, the analysis was done with modern technology at 3D. The text is fluent and clear, and the quality of the images was excellent. The analysis of the data was thorough, and materials and methods including image analysis part were presented accurately and clearly. Ultrastructural analysis was done systematically, and generated models are beautiful and informative. Much thought has put into planning of the experiments and experimental approach. The shortcoming of the study is its limitation to ultrastructural analysis only without attempts to connect to any mechanism. The discussion part contains lot of speculation of the factors that might be needed for the formation and maintenance of the constriction and present several hypotheses for the function of the constriction. The paper would be much stronger if one of few of the leads would be followed, and if there would be any explanation for the role of these structures, or factors affecting them.

Advance:

The paper provides a very nice example for the reuse of publicly archived imaging datasets to complement own experimental work. Hopefully this paper encourages others to the same path, as the large volume EM datasets require significant investments and contain wealth of potential for reuse.

The paper strengthens the description of the ER-NE junction structure significantly and convincingly but does not further our understanding of the mechanisms behind the structure nor the function of them and raises more questions than provides answers. For structural analysis of this kind, the state-of-the-art technology is cryo-EM (e.g., preparation of lamella with cryo-FIB-SEM followed by cryo-tomography), and in this study, the technical limitations come from plastic embedding and ambient temperature imaging. The used techniques would be more adequate for cell biological study, where the described structure is somehow connected to the function in cell, or the factor(s) needed to the formation or maintenance are identified.

Audience:

This study will be of special interest to cell biology community. The study could be an opening to several lines of research, e.g., identification of the factors forming or maintaining the structure, the potential function of the structure, how the structure affects the dynamics of the NE/ER membrane and luminal proteins.

Reviewer's expertise:

The reviewer has long experience in electron microscopy, volume EM techniques and image analysis, and operates mainly in the field of cell biology.

3. How much time do you estimate the authors will need to complete the suggested revisions:

Estimated time to Complete Revisions (Required)

(Decision Recommendation)

Cannot tell / Not applicable

Yes

Revision Plan

Manuscript number: RC-2023-02194

Corresponding author(s): Helena, Bragulat-Teixidor; Shotaro, Otsuka

1. General Statements

The endoplasmic reticulum (ER) and the nucleus are the two largest organelles in eukaryotic cells. The ER membrane is physically connected to the nucleus by junctions with the outer nuclear membrane of the nuclear envelope (NE). The ER-NE junctions are essential for supplying lipids and transmembrane/luminal proteins that are synthesized in the ER to the NE. Here, we revealed that ER-NE junctions are scarce and form constricted hourglass-shaped membranous structures (7–20 nm in diameter) that are significantly distinct from the junctions in the ER in several mammalian cells. We also found that ER-NE junctions become constricted at early telophase, which indicates that their constricted shape is rapidly formed after mitosis and maintained for the rest of the cell cycle. These findings strongly suggest that ER-NE junctions contribute to regulating ER-to-NE transport that affects many important processes including gene expression, nuclear organization, and nuclear pore biogenesis, as well as differentiation, development, and pathogenesis. This work will open lines of investigation into the mechanism and function of this fundamental ER-NE junctions.

2. Description of the planned revisions

Our manuscript has been reviewed by four reviewers through ReviewCommons. Overall, all the reviewers commended the high technical quality of our work and appreciated the impact of our findings as the groundwork for further mechanistic and functional studies.

The reviewers suggested several points to improve our manuscript. The major points are the following: (i) clarify if ER-NE junctions are sheet-sheet or sheet-tubule junctions (Reviewer #2), (ii) quantify the width of ER-ER/NE junctions in late anaphase (Reviewer #2), and (iii) clarify the structural preservation of our sample preparation methods for electron microscopy (Reviewers #3 and #4). The points (i) and (ii) can be addressed by analyzing the EM tomograms that we have already acquired, and the outcome will improve the accuracy and completeness of our manuscript. The point (iii) will be addressed by comparing the morphology of the ER and NE in our EM tomograms with the one in cryo-EM tomograms of rapidly-frozen and FIB-milled cells that are available in public databases. This comparison will clarify the degree of deformation of the sample that our EM protocol might introduce. As outlined in detail below, we will address all the reviewers' comments by additional image analyses, and modifying the text and figures.

Reviewer comments are indicated in *blue italics*, and our response in black.

Revision Plan

Reviewer #1 (Evidence, reproducibility and clarity (Required)):

The mechanisms that differentiate ER from the nuclear envelope (NE) remain to be fully elucidated but likely depend at least in part on junctions between the ER and NE. How such junctions are formed and maintained is the subject of this manuscript where extensive correlative light and electron microscopy is used to observe and characterize ER-nuclear envelope (ER-NE) junctions at distinct phases of the cell cycle. The authors make use of their own electron tomography data as well as publicly available focused-ion beam scanning electron microscopy (FIB-SEM) datasets to compare the morphology of these junctions in different human cell types as well as in budding yeast. The major finding is that ER-NE junctions in human cell lines are more constricted than ER-ER junctions, often to the point of excluding lumen. The examination of mitotic cells suggests that this constriction likely occurs at the end of mitosis as the NE is completing its maturation from ER to NE. The implications of these morphological changes are discussed but there are no mechanistic or functional studies. Overall, the data are well presented, are of high quality and are rigorously evaluated. The manuscript is well written and scholarly, and the speculations as to the function of the constrictions are reasonable. I only have minor comments.

We thank the reviewer for the positive evaluation on our work and for the useful suggestions on how to further improve the manuscript.

1. In Figure 2D, the authors present evidence to demonstrate that an hourglass-like constriction occurs at ER-NE junctions. From the side view, it is difficult to interpret this on the plot, particularly for the ER-NE junctions with a lumen. Perhaps, in the supplemental data, the authors could plot both with and without lumen data separately, and color-code individual traces? I believe this would convey the hourglass nature of these constrictions more clearly.

To make it easier to see individual membrane profiles, we will plot the profiles with and without lumen separately and labelled each profile with distinct colour, as the reviewer suggested.

2. In the Methods section, the authors should describe how carbon-coating of sapphire discs was achieved. If these were provided from the manufacturer pre-coated, this should be specified.

We coated the sapphire discs with carbon by ourselves. We will specify how the carbon-coating was done in the revised manuscript.

3. On page 10, Figure 5F callout 9 lines from the bottom likely should be 5E.

We will correct this error.

Reviewer #1 (Significance (Required)):

Overall, this work provides an important new morphological perspective on the nature of ER-NE junctions in human cells. As the authors describe in their introduction, such junctions have been noted previously in the literature but not in a dedicated study using modern imaging techniques in human cell lines. In describing the morphology of these junctions, the authors lay the groundwork for future mechanistic, functional, and structural studies.

We thank the reviewer for appreciating the significance and the impact of our work.

Revision Plan

Reviewer #2 (Evidence, reproducibility and clarity (Required)):

Summary:

*In this manuscript, Bragulat-Teixidor et al., use correlative live-cell imaging and electron tomography to study the structure of the endoplasmic reticulum-nuclear envelope (ER-NE) junction in HeLa cells (and also in *S. cerevisiae*). The authors also make use of publicly available whole-cell FIB-SEM datasets to study ER-NE junctions in mouse pancreatic islet, HeLa, and human macrophage cells to corroborate their findings in other cell types.*

The authors show that the structure of the ER-NE junction in interphase cells adopts an hourglass shape with a constricted neck. Comparing the ER-NE junction to the ER tubule-sheet junction, the authors show that these structures are different: the ER tubule-sheet junction is not constricted. Because the NE forms from the ER during postmitotic NE assembly, the authors compare the structure of the ER-NE junctions in anaphase, telophase, and interphase cells, and find that the junction becomes constricted in telophase. The number of ER-NE junctions increase going from telophase to interphase.

While the authors do not provide any direct evidence for this, they propose a functional model where the ER-NE junction is constricted because it regulates the supply of certain lipids and proteins from the ER to the NE. One proposed example is that the constriction of the ER-NE junction might prevent the passage of large protein aggregates from entering the NE.

The general question of how the structure of the ER-NE junction might regulate the passage of lipids and proteins from the ER to the NE is interesting and potentially important. However, the authors should address the following issues to improve the accuracy and completeness of this manuscript for it to be considered for publication.

We thank the reviewer for the appreciation of our work and the thoughtful suggestions for further improvements.

Major comments:

1. The authors compare the structure of the ER-NE junction to the structure of the ER tubule-sheet junction in interphase cells. They should instead or in addition be comparing the ER-NE junction to ER sheet-sheet junctions. This is likely a better comparison for two reasons:

i) The NE is similar to an ER sheet due to its flat and extended structure. The ER membranes surrounding the NE consists mostly of a dense network of sheet-like ER (Zheng et al., 2022, PMID: 34912111). Therefore, the ER-NE junction should be compared to these NE-adjacent ER sheet-sheet junctions and not ER tubule-sheet junctions which are likely to be found in the cell periphery.

ii) In HeLa cells, the NE assembles from large ER sheets and not ER tubules (Zhao et al., 2023, PMID:

Revision Plan

37098350; Otsuka et al., 2018, PMID: 29323269; Lu et al., 2011, PMID: 21825076). Therefore, the ER-ER junctions the authors are already studying in anaphase cells are likely to be ER sheet-sheet junctions, which should be kept the same in their analysis of the ER-ER junctions in interphase cells.

Related to this point, comparing the side view panels in Figure 2D with 2H, it seems that the width of the ER membranes on either side of the neck region of the ER-NE junction is in fact getting wider (more sheet-like). This is in contrast to the ER-ER junction where the width stays constant for the ER tubule that is fusing onto the ER sheet. This suggests that indeed, the ER-NE junction is more similar to an ER sheet-sheet junction.

It is a very interesting possibility that the ER-NE junction might be similar to the ER sheet-sheet junction. We will inspect whether the ER that forms the ER-NE junction consists of sheet or tubular ER in our EM tomograms, and describe the outcome in the revised manuscript.

2. The authors claim that in late anaphase cells, the ER-ER/NE (written like this because the ER and NE cannot be distinguished like the authors also point out) junctions are not constricted and had a similar morphology to ER-ER junctions in interphase. However, this claim is only qualitative at the moment, as the authors do not provide any quantification of the width of the ER-ER/NE junctions in late anaphase cells. To make the current claim that the ER-NE junction only becomes constricted in telophase, the authors should report the width of the ER-ER/NE junctions in late anaphase cells.

In late anaphase cells, large ER sheets initially wrap around chromatin at the periphery of the chromosome mass (Zhao et al., 2023, PMID: 37098350; Otsuka et al., 2018, PMID: 29323269; Lu et al., 2011, PMID: 21825076). Therefore, the authors might find it easier to identify ER-ER/NE junctions in the so-called "non-core" regions, instead of in the current regions shown in Figure 3A.

As the reviewer pointed out, we did not provide quantification of the width of ER-ER/NE junctions in late anaphase cells. We will measure them and show the quantification in the revised manuscript.

Minor comments:

1. In the Supplementary Figures 1 A-D, make the scale bars white. Currently, the black scale bars are especially difficult to see in the top panels in Supplementary Figure 1C.

We will change the colour of some scale bars to make them more visible in the Supplementary Figure 1.

2. In the Results section entitled "The number of ER-NE junctions per cell increases from telophase to interphase", the authors should tone down this claim because the number of telophase cells examined is low (only 2 telophase versus 9 interphase cells). It would be better to include the word "slightly" in the title to change it to "slightly increases".

We will modify the text accordingly.

3. In the Results section entitled "The number of ER-NE junctions per cell increases from telophase to interphase", the authors state "These densities were much lower than those of ER-ER junctions...". For sure this is true for ER tubule-tubule junctions in the periphery of the cell as ER tubules form an intricate network by constantly fusing to each other, but it's not clear if this is also the case for ER tubule-sheet or ER sheet-sheet junctions. For clarity, the authors should state that they mean ER tubule-tubule junctions.

Revision Plan

Same comment also for the statement "...although their abundance remains considerably lower than that of ER-ER junctions or nuclear pores at both cell cycle stages". The authors should state that they mean ER tubule-tubule junctions.

We will clarify what we mean by ER-ER junctions in the revised manuscript.

4. In the Results section entitled "The constricted morphology of ER-NE junctions is observed in different mammalian cells, but not in budding yeast", the authors state "...pancreatic islet cells (Figure 5A), HeLa (Figure 5B), and macrophage (Figure 5C) were significantly smaller than most ER-ER junctions (Figure 5F)". The last figure reference here is wrong and should be changed to Figures 5D-E.

We will correct this error.

5. In Discussion, the authors state "Proteins known to form and stabilize junctions in the ER, including Atlastins and Lunapark...". The authors should specify that they mean ER tubule-tubule three-way junctions. Also more generally throughout the manuscript, the authors should be more careful in specifying which ER-ER junctions they mean in each case.

As pointed out in the *Major comment 3* above, we will clarify this point in the revised manuscript.

6. In Discussion, the authors state "Thus, we favour a second scenario in which ER-NE junctions are generated from ER tubules that contact and eventually fuse with the ONM". Given that the ER membranes adjacent to the NE are mostly sheet-like (as pointed out in Major comment 1 above), the authors need to explain how they think an ER tubule (mostly found in the cell periphery) could access and fuse to the NE.

As mentioned in the response to *Major comment 1* above, we will examine if the ER that forms ER-NE junctions is tubule or sheet in our EM tomograms. Depending on the outcome of the examination, we will rephrase the text.

Reviewer #2 (Significance (Required)):

Although the ER-NE junction has been studied in other organisms before, this study represents the first structural characterisation of the ER-NE junction in mammalian cells. Therefore, this study represents an advance for the field in gaining a better understanding of different ER structures and morphologies. How the ER is remodelled during the cell cycle is also an interesting question and an active field of research (Merta et al., 2021 PMID: 34853314; Zhao et al., 2023, PMID: 37098350) which this study further contributes to. This study would therefore be interesting for anyone interested in ER structure/morphology, ER-NE connections, and cell cycle regulation of such ER-NE connections.

My field expertise is in ER and NE. I do not have sufficient expertise to evaluate the methodology for the EM tomography part of this paper.

We thank the reviewer for appreciating the novelty and the impact of our work.

Revision Plan

Reviewer #3 (Evidence, reproducibility and clarity (Required)):

The manuscript by Bragulat-Teixidor et al. is a study of the connection of the ER with the nuclear envelope. It uses advanced ultrastructural techniques: high pressure freezing instead of chemical fixation and EM tomography instead of serial sectioning. Synchronized HeLa cell cultures were examined during interphase, late anaphase (4-6 min after anaphase onset) and early telophase (8-10 minutes after anaphase onset).

The investigators find an unexpected, unusual structure - a constricted neck 7-20 nm wide and about 10 nm long where the ER connects to the nuclear envelope. The 7 nm connections had no apparent lumen. These are not seen in late anaphase when the NE has not yet formed, but they are seen a few minutes later during early telophase when there is a newly formed NE surrounding the chromosomes. A quantitation was made of their abundance, more was found later during interphase, and with wider lumens.

It is very nice to show the EM images as uncolored and segmented (colored). The images shown in the figures are presumably the best that were obtained during the study. Heavy metals do not stain membranes uniformly or exclusively, and identification of structures doesn't always seem unambiguous. The three dimensional information can certainly make this easier though this information is difficult or not possible to show in journal format. In the end, the reader must depend on the judgment of the person who did the analysis. Overall, the analysis seems trustworthy.

We thank the reviewer for the comment. To better present the three-dimensional structure of ER-NE junctions, we will provide movies of the EM sub-tomograms containing the junctions. In this way, the readers will be able to inspect the three-dimensional structure of six ER-NE junctions.

HeLa cells are very convenient for getting information on cell cycle dependence. However, they are cancer cells in culture, so it is important to look at other cell types as well. The same methodology was used on budding yeast and they saw a wide tentlike connection, which reproduces an earlier study. This seems more consistent with what is known or expected from ER membranes. It is not less interesting but perhaps less puzzling.

To get evidence on other mammalian cells, the authors did an analysis of data from OpenOrganelle. These are high pressure frozen cells / tissue imaged by FIB-SEM. The voxels are 4 nm, which is significantly larger than those in EM tomography. Unfortunately, the difficulty of identifying structures is correspondingly more significant. The images shown do not contradict the HeLa results but by themselves (without the HeLa cell data), a convincing case for narrow connections probably couldn't be made.

The reviewer raises a very good point about a limitation of the FIB-SEM datasets in OpenOrganelle. We agree with the reviewer that, as we had mentioned in the manuscript (line 6–11, page 10), the spatial resolution of the FIB-SEM datasets are not enough to gain insights into the exact morphology of the 7–20 nm wide ER-NE junctions because the voxel size is 4 nm. However, the resolution is good enough to examine if ER-NE junctions are narrower than ER-ER junctions, as shown in Figure 5A–E. The fact that

Revision Plan

we rarely found non-constricted ER-NE junctions in FIB-SEM datasets confirms the tiny nature of ER-NE junctions. To clarify this point, we will modify the text (line 24–25 on page 10) as below:

Previous: This analysis of FIB-SEM images confirms the hourglass morphology that distinguishes ER-NE from ER-ER junctions as seen in our EM tomograms...

Revised: This analysis of FIB-SEM images confirms that ER-NE junctions are narrower than ER-ER junctions as seen in our EM tomograms...

The work in this manuscript seems to have been done well. Assuming that this structure is confirmed in other mammalian cells, another kind of question comes to mind: is this the final word on ER to NE connections? The lumenless neck does not seem like it would be a stable structure, somehow it seems like a transient one. In the future, it would help if a new structural protein was identified or some theoretical analysis to help explain the shape.

Certainly, this will not be the final word on ER-NE junctions, which are crucial for the ER-to-NE transport of lipids and transmembrane proteins. In the future, it will be important to identify structural proteins regulating the junctions and reveal how their constricted morphology affects the ER-to-NE transport. We believe that, as you kindly mentioned in the last paragraph of your comments, our observations “serve as a starting point for further structural and functional work” for this unique yet fundamental junctions that connect the ER to nucleus.

It is generally now assumed that high pressure freezing preserves structure perfectly. However, in this reviewer's mind, there is a possibility that some structures are not. The sample is brought to 2000 atmospheres within a few milliseconds, frozen, then the high pressure is released after a second. Although many intracellular structures do seem well preserved, could the junction be susceptible to high pressure? A second source of uncertainty is that in order to embed the samples in resin, the water was removed by freeze substitution. This is known to cause a small amount of tissue shrinkage and possibly could alter a delicate structure. Another way to look at this kind of structure is cryo-EM tomography on hydrated lamellae from plunge frozen cells. I don't recommend that the authors do another arduous, possibly too arduous set of experiments with a completely different technique, but perhaps another group has data which could support their findings.

We think it is very unlikely that ER-NE junctions were deformed due to the high-pressure freezing. In general, high-pressure freezing allows vitrification of specimens up to 0.5 mm in thickness and the vitrification works better for thinner specimens. Our specimens are only 0.02 mm thick monolayer cells frozen in a chamber with 0.03 mm depth. Thus, the vitrification is expected to occur fast and the ER-NE junctions must have been frozen in the same way as in other regions of the cell.

However, as the reviewer pointed out, it is possible that the dehydration of the samples due to freeze substitution might cause deformation in ER-NE junctions. To verify the structural preservation of ER-NE junctions in our protocol, we will compare the morphology of the ER and NE in cryo-EM datasets that are available in public databases with ours. We will describe the outcome in the revised manuscript.

We think that our conclusion from the EM analysis is solid, because we observed significant structural difference between ER-NE junctions and ER-ER junctions in the same cells (Figure 2). In addition, we found the morphology change of ER-NE junctions in late-anaphase, early-telophase, and interphase cells

that were high-pressure frozen and freeze-substituted on the same sapphire disc, and found that the ER-NE junctions became progressively constricted from telophase to interphase (Figure 3).

The following are suggestions for the Discussion:

Yeast have many of the same biochemical processes as mammalian cells. Perhaps their lack of narrow connections can be used as a clue to the function of the narrow necks seen in HeLa cells. For instance, the authors speculate that the narrow connection serves to keep phosphatidylserine in the nuclear envelope low. If the yeast nucleus has the same concentration of phosphatidylserine as the ER, it would provide good evidence for this idea.

Yes, it is indeed the case. It was shown that the yeast outer nuclear membrane has the same concentration of phosphatidylserine as the ER (Tsuji et al., *Proc. Natl. Acad. Sci. U. S. A.*, 2019). We had described this in the discussion on page 14 “this phosphatidylserine enrichment occurs in mammalian cells and not in budding yeast (Tsuji et al., 2019)”, which was probably overlooked by the reviewer. In the revised manuscript, we will rephrase the text to make this point clearer.

There might be other instances of lumenless neck structures. Dynamin mutants can cause a stable constricted tubule - are the dimensions of this tubule similar to that of the ER / NE connections? Or possibly some ESCRT related structure?

These are very interesting questions. As shown in Figure 2A-D and Supplementary Figure 1B, the inner diameter (an inner leaflet distance) of the lumenless ER-NE junctions is below 1 nm. In contrast, the inner diameter of most constricted membrane tubules that the dynamin mutant K44A Dynamin 1 generates is 3.7 nm (Antonny et al., *EMBO J.*, 2016, doi: 10.15252/embj.201694613). The inner diameter of membrane tubules that ESCRT-III subunits CHMP1B and IST1 form is 4.4 nm (Nguyen et al., *Nat. Struct. Mol. Biol.*, 2020, doi: 10.1038/s41594-020-0404-x). Thus, the lumenless ER-NE junctions is unique in their highly-constricted nature and might be regulated by proteins other than dynamin or ESCRT proteins. We will discuss this point in the revised manuscript.

There do not seem to be any recent studies of the ER / nuclear membrane connection in fixed cells. However, there is serial section data online which can be inspected. There are connections in mouse brain cortex in the data of Kasthuri et al., 2015 (<https://neurodata.io/project/ocp/>). Instead of a tubule connection, there seems to be a narrow sheet of ER that connects to the nuclear envelope. But there is something odd about these too. The authors may like to mention something about this or similar work in their manuscript. This reviewer has looked at chemically fixed data from several cell types from his own unpublished data and connections are surprisingly hard to find. Possibly, the connection is particularly sensitive to chemical fixation.

We inspected the serial section data of mouse brain cortex that was chemically fixed. The nuclear envelope in this dataset is deformed and does not seem well preserved. We do not think that we can extract useful information on the ultrastructure of ER-NE junctions from this dataset, and thus will not mention this work in our manuscript.

It is great to hear that the reviewer tried to look for ER-NE junctions in their own EM data. The frequency of ER-NE junctions is rare (only 0.1 junction per square micrometer, Figure 4). Thus, we think that the

Revision Plan

reason why it was hard to find the junctions in the reviewer's data is due to the low-frequency nature of this junction and not due to the chemical fixation.

Reviewer #3 (Significance (Required)):

This is a careful and thorough study of the connection between the ER and the nuclear envelope. The discovery of reticulons and similar proteins, along with biophysical modeling, made the form of the ER accessible to analysis. The factors that govern ER structure are now much better understood. This is particularly true of sheets versus tubules, the three way tubule junctions and to some extent, the junction of ER tubules coming out of the edge of a sheet. However, with all this activity, the subject of the connection of the ER to the nucleus has not been examined in detail. What makes it different is that the tubule is connected perpendicular to the plane of a sheet.

We thank the reviewer for appreciating the quality and novelty of our work.

*The manuscript uses the best ultrastructural techniques and provides strong evidence for a narrow neck at this connection in HeLa cells. With the same methodology, yeast cells (*S. cerevisiae*) have a wider connection. OpenOrganelle data from other mammalian cell types was examined. This data has less resolution and although it does not contradict the HeLa cell data, it does not support it strongly.*

As mentioned in the response to one of this reviewer's comments above, the spatial resolution of FIB-SEM datasets is good enough to examine if ER-NE junctions are narrower than ER-ER junctions. We think that our observation of several mammalian cells in FIB-SEM datasets strongly supports the conclusion that ER-NE junctions are narrower than ER-ER junctions and extends our findings in HeLa cells to two other mammalian cell types.

This work is of interest to cell biologists specializing in membranous organelles or those interested in nuclear physiology. The connection of ER to nuclear envelope is an interesting problem that has not been studied recently. This manuscript could very well serve as a starting point for further structural or functional work by the authors or other groups.

We thank the reviewer for appreciating the significance and impact of our work.

Reviewer #4 (Evidence, reproducibility and clarity (Required)):

Summary:

Membrane bound ribosomes and ER exit sites are present in the cytosolic side of nuclear envelope (NE), suggesting that NE shares protein translocation, folding and quality control functions with the endoplasmic reticulum (ER). Moreover, membrane continuity between the ER and outer NE membrane is evident, and, thus, NE is considered as a subdomain of the ER. To support this, during cell division, NE loses its identity, and participates to daughter cells as part of the ER. However, NE has also membrane proteins and luminal proteins that are enriched to NE and absent from ER during interface, and the

Revision Plan

segregation of NE specific proteins/lipids occurs concomitantly with NE formation during late anaphase/telophase. In this study, the ultrastructure of the ER-NE junctions is described using high resolution electron tomography. Results show convincingly a specific constriction at the ER-NE neck during interface in several mammalian cell types. This structure is absent during metaphase, and also from the budding yeast. Authors present a model for the formation of ER-NE junctions in higher eukaryotes and speculate about their functional role.

We thank the reviewer for the appreciation of our work and the valuable suggestions for further improvements.

Major comments:

The main conclusion of the paper is that although the ER and outer NE membranes are continuous, a specific hourglass shaped constriction at the neck is found in higher mammalian cells during interphase. The structure is specific to ER-NE necks, as it is absent during metaphase and ER-ER junctions. For the analysis, authors used high pressure freezing to ensure best structural preservation. Unfortunately, fixation is not the only potential source of artifacts; during tomography at ambient temperature, the thinning of the plastic sections under the beam can be up to 30%. In evaluation of the results, authors should consider how this thinning could affect the measurements of membrane distances and luminal width, and what type of distortions may happen as a consequence of asymmetric shrinkage.

In addition to analysis of own samples, authors took advantage of the publicly available whole-cell datasets in OpenOrganelle and used these datasets to expand the number of cell types analyzed.

Moreover, the 3D-datasets were generated with different imaging technique, FIB-SEM. Although this technique provides lower resolution in general, it provides isotropic resolution, and the data could be used to eliminate the shortcomings of the tomography, thinning of the sections and the missing wedge.

The authors could expand the comparison of the data from these different sources from this perspective, especially since HeLa cells were used in their own tomography studies and FIB-SEM datasets in OpenOrganelle. Similarly, it would be interesting to see if similar approach could be used to compare their results to those obtained by cryo-EM by utilizing the cryo-EM database. Have authors checked if any suitable datasets for analysis of ER-NE junctions could be found from public archives?

For the analysis of mitotic cells, double thymidine block was used to synchronize the cell culture. It is not clear, why synchronization was necessary, as CLEM was used to select the cells, and their number was rather low. Do cells continue growing and synthesizing new proteins during thymidine blocks? As one way to control potential artifacts due to the synchronization treatment, authors could compare the average thickness of ER and NE in naturally occurring interphase and mitotic cells vs. synchronized cells.

We agree with the reviewer that it is important to clarify the degree of shrinkage and deformation of the sample that our EM protocol might introduce. To assess the degree of sample shrinkage and deformation in the plastic sections, we will compare the ONM-INM distance measured in our plastic sections with the one in cryo-EM tomograms of rapidly-frozen and FIB-milled mammalian cells that are publically available (EMPIAR, the Electron Microscopy Public Image Archive, <https://www.ebi.ac.uk/pdbe/emdb/empiar/>), and describe the outcome in the revised manuscript.

The reason why we synchronized the cell cycle is to enrich cells in late anaphase and early telophase in the same plastic sections, so that we can compare their ultrastructure side-by-side. In the revised

Revision Plan

manuscript, we will examine if the double thymidine block affects the ER-NE junction morphology by comparing the morphology of the ER and NE between the synchronised and non-synchronised cells.

As we described in the response to Reviewer 3, we think that our conclusion from the EM analysis is solid because of the following reasons. (i) We observed a significant structural difference between ER-NE junctions and ER-ER junctions in the same cells (Figure 2). (ii) We discovered a morphology change of ER-NE junctions in late-anaphase, early-telophase, and interphase cells that were freeze-substituted on the same sapphire disc; the ER-NE junctions became progressively constricted from telophase to interphase (Figure 3).

Minor comments:

On page 5, last chapter (+ Fig.1 legend and materials and methods): "the quick tomograms covered the entire NE" is misleading, as the imaging covered a thin layer of the entire NE only. - Authors could have analyzed the entire NE from the FIB-SEM datasets but chose to use stereological approach to minimize their work.

We will modify the text to make it clear that the quick tomograms covered the NE in a section and not the entire NE of the cell in the revised manuscript.

To save time from the readers to follow the reference, authors could describe how the specimens used in OpenOrganelle datasets were fixed and processed, especially as they emphasize the importance of high pressure freezing in their own sample prep. Similarly, in Fig.4 legend, authors refer to measurements done in the previous study without explaining how and from what type of data.

We thank the reviewer for pointing these out. We will describe how the OpenOrganelle datasets were generated and how the nuclear surface area measurement was done.

Is there a difference between mesh generation and segmentation, or is it just two different terms used for the same thing by different programs?

We apologize our short description of these terms. We will clarify these terms in the revised manuscript.

Reviewer #4 (Significance (Required)):

General assessment:

ER-NE gates were described earlier in the literature for specific cell types using standard thin-section TEM imaging, and in this study, the analysis was done with modern technology at 3D. The text is fluent and clear, and the quality of the images was excellent. The analysis of the data was thorough, and materials and methods including image analysis part were presented accurately and clearly.

Ultrastructural analysis was done systematically, and generated models are beautiful and informative. Much thought has put into planning of the experiments and experimental approach.

The shortcoming of the study is its limitation to ultrastructural analysis only without attempts to connect to any mechanism. The discussion part contains lot of speculation of the factors that might be needed for the formation and maintenance of the constriction and present several hypotheses for the function of the constriction. The paper would be much stronger if one of few of the leads would be followed, and if there

Revision Plan

would be any explanation for the role of these structures, or factors affecting them.

We thank the reviewer for the appreciation of the clarity and quality of our work. The molecular mechanism that regulates the function, shape and biogenesis of ER-NE junctions will be the subject of future studies, for which our discovery of a highly-constricted morphology of the ER-NE junctions lays the groundwork.

Advance:

The paper provides a very nice example for the reuse of publicly archived imaging datasets to complement own experimental work. Hopefully this paper encourages others to the same path, as the large volumeEM datasets require significant investments and contain wealth of potential for reuse.

We strongly agree with the reviewer. The volume EM datasets that are publically available contain wealth of potential for new discoveries. We also hope that our paper encourages other scientists to make good use of those datasets and also to deposit their own data to the public databases. We will deposit our EM tomograms to EMPIAR, the Electron Microscopy Public Image Archive.

The paper strengthens the description of the ER-NE junction structure significantly and convincingly but does not further our understanding of the mechanisms behind the structure nor the function of them and raises more questions than provides answers. For structural analysis of this kind, the state-of-the-art technology is cryo-EM (e.g., preparation of lamella with cryo-FIB-SEM followed by cryo-tomography), and in this study, the technical limitations come from plastic embedding and ambient temperature imaging. The used techniques would be more adequate for cell biological study, where the described structure is somehow connected to the function in cell, or the factor(s) needed to the formation or maintenance are identified.

Indeed, a limitation of our current study is that we did not reveal the underlying molecular mechanism and the functions of the constricted morphology of ER-NE junctions. We do not think that cryo-EM is necessarily required because we have collected evidence that the ER-NE connections are distinct from the ER-ER junctions in not only our EM tomography data (Fig. 2) but also in the EM datasets deposited in public databases (Fig. 5).

Audience:

This study will be of special interest to cell biology community. The study could be an opening to several lines of research, e.g., identification of the factors forming or maintaining the structure, the potential function of the structure, how the structure affects the dynamics of the NE/ER membrane and luminal proteins.

We thank the reviewer for appreciating the impact of our work.

Reviewer's expertise:

The reviewer has long experience in electron microscopy, volumeEM techniques and image analysis, and operates mainly in the field of cell biology.

Revision Plan

3. Description of the revisions that have already been incorporated in the transferred manuscript

4. Description of analyses that authors prefer not to carry out

Dear Shotaro,

Thank you for the transfer of your research manuscript to our journal. As discussed, we would like to invite you to revise your manuscript as outlined in your point-by-point response to the referees from Review Commons.

Please address all referee concerns in a complete point-by-point response. Acceptance of the manuscript will depend on a positive outcome of a second round of review. It is EMBO Reports policy to allow a single round of revision only and acceptance or rejection of the manuscript will therefore depend on the completeness of your responses included in the next, final version of the manuscript.

We realize that it is difficult to revise to a specific deadline. In the interest of protecting the conceptual advance provided by the work, we recommend a revision within 3 months (March 15). Please discuss the revision progress ahead of this time with the editor if you require more time to complete the revisions.

I am also happy to discuss the revision further via e-mail or a video call, if you wish.

*****IMPORTANT NOTE:

We perform an initial quality control of all revised manuscripts before re-review. Your manuscript will FAIL this control and the handling will be delayed IN CASE the following APPLIES:

- 1) A data availability section providing access to data deposited in public databases is missing. If you have not deposited any data, please add a sentence to the data availability section that explains that.
- 2) Your manuscript contains statistics and error bars based on $n=2$. Please use scatter blots in these cases. No statistics should be calculated if $n=2$.

When submitting your revised manuscript, please carefully review the instructions that follow below. Failure to include requested items will delay the evaluation of your revision. *****

- 1) a .docx formatted version of the manuscript text (including legends for main figures, EV figures and tables). Please make sure that the changes are highlighted to be clearly visible.
- 2) individual production quality figure files as .eps, .tif, .jpg (one file per figure).
Please download our Figure Preparation Guidelines (figure preparation pdf) from our Author Guidelines pages <https://www.embopress.org/page/journal/14693178/authorguide> for more info on how to prepare your figures.
- 3) a .docx formatted letter INCLUDING the reviewers' reports and your detailed point-by-point responses to their comments. As part of the EMBO Press transparent editorial process, the point-by-point response is part of the Review Process File (RPF), which will be published alongside your paper.
- 4) a complete author checklist, which you can download from our author guidelines (). Please insert information in the checklist that is also reflected in the manuscript. The completed author checklist will also be part of the RPF.
- 5) Please note that all corresponding authors are required to supply an ORCID ID for their name upon submission of a revised manuscript (). Please find instructions on how to link your ORCID ID to your account in our manuscript tracking system in our Author guidelines
()
- 6) We replaced Supplementary Information with Expanded View (EV) Figures and Tables that are collapsible/expandable online. A maximum of 5 EV Figures can be typeset. EV Figures should be cited as 'Figure EV1, Figure EV2' etc... in the text and their respective legends should be included in the main text after the legends of regular figures.

- Additional Tables/Datasets should be labeled and referred to as Table EV1, Dataset EV1, etc. Legends have to be provided in

a separate tab in case of .xls files. Alternatively, the legend can be supplied as a separate text file (README) and zipped together with the Table/Dataset file.

7) Please note that a Data Availability section at the end of Materials and Methods is now mandatory. In case you have no data that requires deposition in a public database, please state so instead of refereeing to the database. See also < <https://www.embopress.org/page/journal/14693178/authorguide#dataavailability>>. Please note that the Data Availability Section is restricted to new primary data that are part of this study.

Additional information on source data and instruction on how to label the files are available .

10) Figure legends and data quantification:
The following points must be specified in each figure legend:

- the name of the statistical test used to generate error bars and P values,
- the number (n) of independent experiments (please specify technical or biological replicates) underlying each data point,
- the nature of the bars and error bars (s.d., s.e.m.)

- If the data are obtained from n {less than or equal to} 5, show the individual data points in addition to the SD or SEM.
- If the data are obtained from n {less than or equal to} 2, use scatter blots showing the individual data points.

See also the guidelines for figure legend preparation:
<https://www.embopress.org/page/journal/14693178/authorguide#figureformat>

11) Our journal encourages inclusion of *data citations in the reference list* to directly cite datasets that were re-used and obtained from public databases. Data citations in the article text are distinct from normal bibliographical citations and should directly link to the database records from which the data can be accessed. In the main text, data citations are formatted as follows: "Data ref: Smith et al, 2001" or "Data ref: NCBI Sequence Read Archive PRJNA342805, 2017". In the Reference list, data citations must be labeled with "[DATASET]". A data reference must provide the database name, accession number/identifiers and a resolvable link to the landing page from which the data can be accessed at the end of the reference. Further instructions are available at .

12) All Materials and Methods need to be described in the main text. We would encourage you to use 'Structured Methods', our new Materials and Methods format. According to this format, the Materials and Methods section should include a Reagents and Tools Table (listing key reagents, experimental models, software and relevant equipment and including their sources and relevant identifiers) followed by a Methods and Protocols section in which we encourage the authors to describe their methods using a step-by-step protocol format with bullet points, to facilitate the adoption of the methodologies across labs.

More information on how to adhere to this format as well as downloadable templates (.doc or .xls) for the Reagents and Tools Table can be found in our author guidelines: < <https://www.embopress.org/page/journal/14693178/authorguide#manuscriptpreparation>>. An example of a Method paper with Structured Methods can be found here: .

13) As part of the EMBO publication's Transparent Editorial Process, EMBO Reports publishes online a Review Process File to accompany accepted manuscripts. This File will be published in conjunction with your paper and will include the referee reports, your point-by-point response and all pertinent correspondence relating to the manuscript.

Kind regards,

Martina

Reviewer comments are indicated in *blue italics*, and our response in black.

Reviewer #1 (Evidence, reproducibility and clarity (Required)):

The mechanisms that differentiate ER from the nuclear envelope (NE) remain to be fully elucidated but likely depend at least in part on junctions between the ER and NE. How such junctions are formed and maintained is the subject of this manuscript where extensive correlative light and electron microscopy is used to observe and characterize ER-nuclear envelope (ER-NE) junctions at distinct phases of the cell cycle. The authors make use of their own electron tomography data as well as publicly available focused-ion beam scanning electron microscopy (FIB-SEM) datasets to compare the morphology of these junctions in different human cell types as well as in budding yeast. The major finding is that ER-NE junctions in human cell lines are more constricted than ER-ER junctions, often to the point of excluding lumen. The examination of mitotic cells suggests that this constriction likely occurs at the end of mitosis as the NE is completing its maturation from ER to NE. The implications of these morphological changes are discussed but there are no mechanistic or functional studies. Overall, the data are well presented, are of high quality and are rigorously evaluated. The manuscript is well written and scholarly, and the speculations as to the function of the constrictions are reasonable. I only have minor comments.

We thank the reviewer for the positive evaluation on our work and for the useful suggestions on how to further improve the manuscript.

1. In Figure 2D, the authors present evidence to demonstrate that an hourglass-like constriction occurs at ER-NE junctions. From the side view, it is difficult to interpret this on the plot, particularly for the ER-NE junctions with a lumen. Perhaps, in the supplemental data, the authors could plot both with and without lumen data separately, and color-code individual traces? I believe this would convey the hourglass nature of these constrictions more clearly.

To make it easier to see individual membrane profiles, we have plotted the profiles with and without lumen separately and labelled each profile with distinct colour, as the reviewer suggested. We now show these new plots in the new Figure EV1B.

2. In the Methods section, the authors should describe how carbon-coating of sapphire discs was achieved. If these were provided from the manufacturer precoated, this should be specified.

We coated the sapphire discs with carbon by ourselves. We have specified how the carbon-coating was done in the Methods section on page 17.

3. On page 10, Figure 5F callout 9 lines from the bottom likely should be 5E.

We have corrected this error.

Reviewer #1 (Significance (Required)):

Overall, this work provides an important new morphological perspective on the nature of ER-NE junctions in human cells. As the authors describe in their introduction, such junctions have been noted previously in the literature but not in a dedicated study using modern imaging techniques in human cell lines. In describing the morphology of these junctions, the authors lay the groundwork for future mechanistic, functional, and structural studies.

We thank the reviewer for appreciating the significance and the impact of our work.

Reviewer #2 (Evidence, reproducibility and clarity (Required)):

Summary:

*In this manuscript, Bragulat-Teixidor et al., use correlative live-cell imaging and electron tomography to study the structure of the endoplasmic reticulum-nuclear envelope (ER-NE) junction in HeLa cells (and also in *S. cerevisiae*). The authors also make use of publicly available whole-cell FIB-SEM datasets to study ER-NE junctions in mouse pancreatic islet, HeLa, and human macrophage cells to corroborate their findings in other cell types.*

The authors show that the structure of the ER-NE junction in interphase cells adopts an hourglass shape with a constricted neck. Comparing the ER-NE junction to the ER tubule-sheet junction, the authors show that these structures are different: the ER tubule-sheet junction is not constricted. Because the NE forms from the ER during postmitotic NE assembly, the authors compare the structure of the ER-NE junctions in anaphase, telophase, and interphase cells, and find that the junction becomes constricted in telophase. The number of ER-NE junctions increase going from telophase to interphase.

While the authors do not provide any direct evidence for this, they propose a functional model where the ER-NE junction is constricted because it regulates the supply of certain lipids and proteins from the ER to the NE. One proposed example is that the constriction of the ER-NE junction might prevent the passage of large protein aggregates from entering the NE.

The general question of how the structure of the ER-NE junction might regulate the passage of lipids and proteins from the ER to the NE is interesting and potentially important. However, the authors should address the following issues to improve the accuracy and completeness of this manuscript for it to be considered for publication.

We thank the reviewer for the appreciation of our work and the thoughtful suggestions for further improvements.

Major comments:

1. The authors compare the structure of the ER-NE junction to the structure of the ER tubule-sheet junction in interphase cells. They should instead or in addition be comparing the ER-NE junction to ER sheet-sheet junctions. This is likely a better comparison for two reasons:

i) The NE is similar to an ER sheet due to its flat and extended structure. The ER membranes surrounding the NE consists mostly of a dense network of sheet-like ER (Zheng et al., 2022, PMID: 34912111). Therefore, the ER-NE junction should be compared to these NE-adjacent ER sheet-sheet junctions and not ER tubule-sheet junctions which are likely to be found in the cell periphery.

We thank the reviewer for pointing this out and apologize for our short description of the definition of ER junctions. We have now provided new supplementary figures showing (i) the ER junctions that we analysed are mostly perinuclear (new Figure EV1E) and (ii) the shape of the approaching ER to a flat ER sheet is mostly sheet like at 100 nm away from the junction base (new Figure EV1C). In the previous version of the manuscript, we referred to these junctions as “ER tubule-sheet junction” because the cross-section of the approaching ER was considerably smaller than the flat sheet base. To avoid such confusion, we have revised the text on pages 7-8 and 21 as below:

Previous: We analysed specifically junctions that had a similar topology to ER–NE junctions, i.e. an ER tubule fused perpendicularly to an ER sheet.

Revised: We analysed specifically the ER–ER junctions that are located near the NE and had a similar topology to ER–NE junctions, i.e. an ER piece fused to the flat side of an ER sheet.

ii) In HeLa cells, the NE assembles from large ER sheets and not ER tubules (Zhao et al., 2023, PMID: 37098350; Otsuka et al., 2018, PMID: 29323269; Lu et al., 2011, PMID: 21825076).

Therefore, the ER-ER junctions the authors are already studying in anaphase cells are likely to be ER sheet-sheet junctions, which should be kept the same in their analysis of the ER-ER junctions in interphase cells.

We have now quantified the shape of the ER-ER/NE junctions in anaphase cells. We noticed that, at most of the ER-ER/NE junctions, the ER pieces connect to the edge of flat membrane sheets and not to the flat side of sheets (new Figures 3A, EV4A). The topology of ER-ER/NE junctions in anaphase is not comparable to the ER-ER junctions in interphase, and thus we have decided not to compare their morphology to ER-ER junctions in interphase. Instead, we focus on the fact that the ER-ER/NE junctions in anaphase are much wider and elongated than the ones in telophase and interphase. We have clarified this point in the manuscript (page 8-9).

Related to this point, comparing the side view panels in Figure 2D with 2H, it seems that the width of the ER membranes on either side of the neck region of the ER-NE junction is in fact getting wider (more sheet-like). This is in contrast to the ER-ER junction where the width stays constant for the ER tubule that is fusing onto the ER sheet. This suggests that indeed, the ER-NE junction is more similar to an ER sheet-sheet junction.

To clarify this point, we have now examined whether the ER piece that forms the ER-NE junction is tubule- or sheet-like ER at 100 nm away from the junction base. As shown in the new Figure EV1C, the ER that forms ER-NE junctions is a mixture of tubules, sheets, and with other morphologies. The result indicates that the ER-NE junctions are constricted irrespective of the shape of the approaching ER. Moreover, we captured one example in which the ER piece forms simultaneously a constricted junction to the NE and a wide junction to an ER sheet (new Figure EV1F, Movie EV4), which highlights that the constricted hourglass shape is an exclusive feature to ER-NE junctions. We have included these new analyses in the revised manuscript. We have also provided videos that show the 3D EM images and meshes of these junctions, so that anyone can evaluate the shape of the approaching ER (new Movies EV2–6).

2. The authors claim that in late anaphase cells, the ER-ER/NE (written like this because the ER and NE cannot be distinguished like the authors also point out) junctions are not constricted and had a similar morphology to ER-ER junctions in interphase. However, this claim is only qualitative at the moment, as the authors do not provide any quantification of the width of the ER-ER/NE junctions in late anaphase cells. To make the current claim that the ER-NE junction only becomes constricted in telophase, the authors should report the width of the ER-ER/NE junctions in late anaphase cells.

In late anaphase cells, large ER sheets initially wrap around chromatin at the periphery of the chromosome mass (Zhao et al., 2023, PMID: 37098350; Otsuka et al., 2018, PMID: 29323269; Lu et al., 2011, PMID: 21825076). Therefore, the authors might find it easier to identify ER-ER/NE junctions in the so-called "non-core" regions, instead of in the current regions shown in Figure 3A. We thank the reviewer for the constructive suggestion. We have now quantified the 3D ultrastructure of the ER-ER/NE junctions in late anaphase cells. As shown in the new Figures 3 and EV2E,F, the width and aspect ratio of the ER-NE junctions in late anaphase are indeed larger and distinct from the ER-NE junctions in early telophase and interphase. We have added these new data to the revised manuscript.

Regarding the comparison of ER-ER/NE junctions in late anaphase with ER-ER junctions in interphase, we do not compare their morphology as we described in the response to your previous comment and focus on the fact that the ER-ER/NE junctions in anaphase are much wider and elongated than the ones in telophase and interphase. We have modified two parts in the main text on pages 8 and 9 as below:

Previous: The junctions at these ER membranes touching chromatin were not constricted and had a similar morphology to ER-ER junctions in interphase.

Revised: At these ER membrane junctions touching chromatin, the ER pieces connect to the edge of flat membrane sheets and not to the flat side of sheets. The junctions were not constricted and were

significantly wider and more elongated than the ER–NE junctions in interphase (Figure 3A,B, Figure EV2E,F).

Previous: Ultrastructural analysis revealed that most of ER–NE junctions in telophase... The average width... is between that of ER–NE and ER–ER junctions in interphase...

Revised: Ultrastructural analysis revealed that most ER–NE junctions in telophase... The average width... is between ER–ER/NE junctions in anaphase and ER–NE junctions in interphase.

Minor comments:

1. In the Supplementary Figures 1 A-D, make the scale bars white. Currently, the black scale bars are especially difficult to see in the top panels in Supplementary Figure 1C.

We have positioned the scale bars outside the images to make them more visible in the new Figure EV1A (previous Supplementary Figure 1A-D).

2. In the Results section entitled "The number of ER-NE junctions per cell increases from telophase to interphase", the authors should tone down this claim because the number of telophase cells examined is low (only 2 telophase versus 9 interphase cells). It would be better to include the word "slightly" in the title to change it to "slightly increases".

We have modified the text accordingly on page 9.

3. In the Results section entitled "The number of ER-NE junctions per cell increases from telophase to interphase", the authors state "These densities were much lower than those of ER-ER junctions...". For sure this is true for ER tubule-tubule junctions in the periphery of the cell as ER tubules form an intricate network by constantly fusing to each other, but it's not clear if this is also the case for ER tubule-sheet or ER sheet-sheet junctions. For clarity, the authors should state that they mean ER tubule-tubule junctions.

Same comment also for the statement "...although their abundance remains considerably lower than that of ER-ER junctions or nuclear pores at both cell cycle stages". The authors should state that they mean ER tubule-tubule junctions

We apologize our short description of the ER-ER junctions. What we mean by “ER-ER junctions” here is all types of junctions; i.e., when the ER is observed by electron microscopy at nanometer resolution in a whole cell, the ER is interconnect by junctions of tubule-tubule, tubule to the edge of sheet, tubule to the flat side of sheet, sheet-sheet, fenestrated sheets, etc. (Heinrich et al., Nature, 2021, DOI: 10.1038/s41586-021-03977-3; Parlakgöl et al., Nature, 2022, DOI: 10.1038/s41586-022-04488-5). To clarify that we refer to all types of junctions within the ER network, we have specified it as “junctions within the highly interconnected ER network of sheets and tubules (Heinrich et al., 2021; Tikhomirova et al., 2022)” on page 10.

4. In the Results section entitled "The constricted morphology of ER-NE junctions is observed in different mammalian cells, but not in budding yeast", the authors state "...pancreatic islet cells (Figure 5A), HeLa (Figure 5B), and macrophage (Figure 5C) were significantly smaller than most ER-ER junctions (Figure 5F)". The last figure reference here is wrong and should be changed to Figures 5D-E.

We have corrected this error.

5. In Discussion, the authors state "Proteins known to form and stabilize junctions in the ER, including Atlantins and Lunapark...". The authors should specify that they mean ER tubule-tubule three-way junctions. Also more generally throughout the manuscript, the authors should be more careful in specifying which ER-ER junctions they mean in each case.

We have clarified that the ER junctions what we mean here are three-way tubular junctions in the revised manuscript (on page 13). We have also specified which ER-ER junctions we mean in other parts of the text (on pages 7, 8, 12).

6. In Discussion, the authors state "Thus, we favour a second scenario in which ER-NE junctions are generated from ER tubules that contact and eventually fuse with the ONM". Given that the ER membranes adjacent to the NE are mostly sheet-like (as pointed out in Major comment 1 above), the authors need to explain how they think an ER tubule (mostly found in the cell periphery) could access and fuse to the NE.

As mentioned in the response to *Major comment 1* above, the ER pieces that form ER-NE junctions are mixtures of tubules and sheets. Therefore, we have modified the text on page 15 as below:

Previous: ...ER-NE junctions are generated from ER tubules...

Revised: ...ER-NE junctions are generated from ER tubules or sheets...

Reviewer #2 (Significance (Required)):

Although the ER-NE junction has been studied in other organisms before, this study represents the first structural characterisation of the ER-NE junction in mammalian cells. Therefore, this study represents an advance for the field in gaining a better understanding of different ER structures and morphologies. How the ER is remodelled during the cell cycle is also an interesting question and an active field of research (Merta et al., 2021 PMID: 34853314; Zhao et al., 2023, PMID: 37098350) which this study further contributes to. This study would therefore be interesting for anyone interested in ER structure/morphology, ER-NE connections, and cell cycle regulation of such ER-NE connections.

My field expertise is in ER and NE. I do not have sufficient expertise to evaluate the methodology for the EM tomography part of this paper.

We thank the reviewer for appreciating the novelty and the impact of our work.

Reviewer #3 (Evidence, reproducibility and clarity (Required)):

The manuscript by Bragulat-Teixidor et al. is a study of the connection of the ER with the nuclear envelope. It uses advanced ultrastructural techniques: high pressure freezing instead of chemical fixation and EM tomography instead of serial sectioning. Synchronized HeLa cell cultures were examined during interphase, late anaphase (4-6 min after anaphase onset) and early telophase (8-10 minutes after anaphase onset).

The investigators find an unexpected, unusual structure - a constricted neck 7-20 wide and about 10 nm long where the ER connects to the nuclear envelope. The 7 nm connections had no apparent lumen. These are not seen in late anaphase when the NE has not yet formed, but they are seen a few minutes later during early telophase when there is a newly formed NE surrounding the chromosomes. A quantitation was made of their abundance, more was found later during interphase, and with wider lumens.

It is very nice to show the EM images as uncolored and segmented (colored). The images shown in the figures are presumably the best that were obtained during the study. Heavy metals do not stain membranes uniformly or exclusively, and identification of structures doesn't always seem unambiguous. The three dimensional information can certainly make this easier though this

information is difficult or not possible to show in journal format. In the end, the reader must depend on the judgment of the person who did the analysis. Overall, the analysis seems trustworthy.

We thank the reviewer for the comment. To better convey the three-dimensional structure of ER-NE junctions, we provided movies of the EM sub-tomograms containing the junctions. In the new Movies EV2–6, the reader can inspect the three-dimensional structure of six more ER-NE junctions. In addition, we have deposited the tomograms containing junctions to the Electron Microscopy Public Image Archive (EMPIAR; <https://www.ebi.ac.uk/pdbe/emdb/empiar/>) under accession codes 12025, 12048, 12050, 12051, so that anyone can observe and judge the ultrastructure of the junctions. The tomograms will be available after the publication of this manuscript, but can be accessed for reviewers by these credentials: *information redacted*.

HeLa cells are very convenient for getting information on cell cycle dependence. However, they are cancer cells in culture, so it is important to look at other cell types as well. The same methodology was used on budding yeast and they saw a wide tentlike connection, which reproduces an earlier study. This seems more consistent with what is known or expected from ER membranes. It is not less interesting but perhaps less puzzling.

To get evidence on other mammalian cells, the authors did an analysis of data from OpenOrganelle. These are high pressure frozen cells / tissue imaged by FIB-SEM. The voxels are 4 nm, which is significantly larger than those in EM tomography. Unfortunately, the difficulty of identifying structures is correspondingly more significant. The images shown do not contradict the HeLa results but by themselves (without the HeLa cell data), a convincing case for narrow connections probably couldn't be made.

The reviewer raises a very good point about a limitation of the FIB-SEM datasets in OpenOrganelle. We agree with the reviewer that, as we had mentioned in the manuscript (page 10), the spatial resolution of the FIB-SEM datasets is not enough to gain insights into the exact morphology of the 7–20 nm wide ER-NE junctions because the voxel size is 4 nm. However, the resolution is good enough to examine if ER-NE junctions are narrower than ER-ER junctions, as shown in Figure 5A–E. The fact that we rarely found non-constricted ER-NE junctions in FIB-SEM datasets confirms the tiny nature of ER-NE junctions in other mammalian cells. To clarify this point, we have modified the text on page 11 as below:

Previous: This analysis of FIB-SEM images confirms the hourglass morphology that distinguishes ER–NE from ER–ER junctions as seen in our EM tomograms...

Revised: This analysis of FIB-SEM images confirms that ER-NE junctions are narrower than ER-ER junctions as seen in our EM tomograms...

The work in this manuscript seems to have been done well. Assuming that this structure is confirmed in other mammalian cells, another kind of question comes to mind: is this the final word on ER to NE connections? The lumenless neck does not seem like it would be a stable structure, somehow it seems like a transient one. In the future, it would help if a new structural protein was identified or some theoretical analysis to help explain the shape.

Certainly, this will not be the final word on ER-NE junctions, which we think are crucial for the ER-to-NE transport of lipids and transmembrane proteins. In the future, it will be important to identify structural proteins regulating the junctions and reveal how their constricted morphology affects the ER-to-NE transport. We believe that, as you kindly mentioned in the last paragraph of your comments, our observations “serve as a starting point for further structural and functional work” for this unique yet fundamental junctions that connect the ER to nucleus.

It is generally now assumed that high pressure freezing preserves structure perfectly. However, in this reviewer's mind, there is a possibility that some structures are not. The sample is brought to 2000 atmospheres within a few milliseconds, frozen, then the high pressure is released after a second. Although many intracellular structures do seem well preserved, could the junction be susceptible to high pressure? A second source of uncertainty is that in order to embed the samples in resin, the water

was removed by freeze substitution. This is known to cause a small amount of tissue shrinkage and possibly could alter a delicate structure. Another way to look at this kind of structure is cryo-EM tomography on hydrated lamellae from plunge frozen cells. I don't recommend that the authors do another arduous, possibly too arduous set of experiments with a completely different technique, but perhaps another group has data which could support their findings.

We think it is very unlikely that ER-NE junctions were deformed due to the high-pressure freezing. In general, high-pressure freezing allows vitrification of specimens up to 0.5 mm in thickness and the vitrification works better for thinner specimens. Our specimens are only 0.02 mm thick monolayer cells frozen in a chamber with 0.025 mm depth. Thus, the vitrification is expected to occur fast and the ER-NE junctions must have been frozen in the same way as other regions of the cell.

However, as the reviewer pointed out, it is possible that the dehydration of the samples due to freeze substitution might cause deformation in ER-NE junctions. To verify the structural preservation of ER-NE junctions in our protocol, we looked for ER-NE junctions in cryo-EM tomograms of HeLa cells that are publicly available on EMPIAR (the Electron Microscopy Public Image Archive). Unfortunately, we could not find ER-NE junctions in those datasets. Given that the frequency of ER-NE junctions is very low (~0.13 junction per square micrometer, Figure 4C), more cryo-EM tomograms of the NE would need to be taken and inspected to find them. Thus, we cannot technically exclude the possibility that ER-NE junctions are deformed due to the freeze substitution. However, we think that the conclusion from our EM analysis is solid because we observed structural differences between ER-NE and ER-ER junctions in the same cells (Figure 2). In addition, we inspected ER-NE junctions in late-anaphase, early-telophase, and interphase cells that were high-pressure frozen and freeze-substituted on the same region of the sapphire disc, and found that the ER-NE junctions became progressively constricted from telophase to interphase (Figure 3). Furthermore, we did not observe highly constricted ER-NE junctions in yeast cells that we high-pressure froze and freeze-substituted into resin in the same way as HeLa cells. Therefore, we think that the highly-constricted shape of ER-NE junctions in interphase mammalian cells is not an artefact but rather represents the native structure.

The following are suggestions for the Discussion:

Yeast have many of the same biochemical processes as mammalian cells. Perhaps their lack of narrow connections can be used as a clue to the function of the narrow necks seen in HeLa cells. For instance, the authors speculate that the narrow connection serves to keep phosphatidylserine in the nuclear envelope low. If the yeast nucleus has the same concentration of phosphatidylserine as the ER, it would provide good evidence for this idea.

Yes, it is indeed the case. It was shown that the yeast outer nuclear membrane has the same concentration of phosphatidylserine as the ER does (Tsuji et al., *Proc. Natl. Acad. Sci. U. S. A.*, 2019). We had described this in the discussion on page 15 “this phosphatidylserine enrichment occurs in mammalian cells and not in budding yeast (Tsuji et al., 2019)”.

There might be other instances of lumenless neck structures. Dynamin mutants can cause a stable constricted tubule - are the dimensions of this tubule similar to that of the ER / NE connections? Or possibly some ESCRT related structure?

These are very interesting questions. As shown in Figure 2A-D, the inner diameter (an inner leaflet distance) of the lumenless ER-NE junctions is below 1 nm. In contrast, the inner diameter of most constricted membrane tubules that the dynamin mutant (K44A Dynamin 1) generates is 3.7 nm (Antonny et al., *EMBO J.*, 2016, doi: 10.15252/embj.201694613). The inner diameter of membrane tubules that ESCRT-III subunits CHMP1B and IST1 create is 4.4 nm (Nguyen et al., *Nat. Struct. Mol. Biol.*, 2020, doi: 10.1038/s41594-020-0404-x). Thus, the lumenless ER-NE junctions is unique in their highly-constricted nature and might be regulated by proteins other than dynamin or ESCRT proteins. We have included this point in the Discussion on page 13-14.

There do not seem to be any recent studies of the ER / nuclear membrane connection in fixed cells.

However, there is serial section data online which can be inspected. There are connections in mouse brain cortex in the data of Kasthuri et al., 2015 (<https://neurodata.io/project/ocp/>). Instead of a tubule connection, there seems to be a narrow sheet of ER that connects to the nuclear envelope. But there is something odd about these too. The authors may like to mention something about this or similar work in their manuscript. This reviewer has looked at chemically fixed data from several cell types from his own unpublished data and connections are surprisingly hard to find. Possibly, the connection is particularly sensitive to chemical fixation.

We inspected the serial section data of mouse brain cortex that was chemically fixed. The nuclear envelope in this dataset is deformed and does not seem well preserved. We do not think that we can extract useful information on the ultrastructure of ER-NE junctions from this dataset, and thus will not mention this work in our manuscript.

It is great to hear that the reviewer tried to look for ER-NE junctions in their own EM data. The frequency of ER-NE junctions is rare (only 0.13 junction per square micrometer, Figure 4C). We think that the reason why it was hard to find the junctions in the reviewer's data is due to the low-frequency nature of this junction and not due to the chemical fixation.

Reviewer #3 (Significance (Required)):

This is a careful and thorough study of the connection between the ER and the nuclear envelope. The discovery of reticulons and similar proteins, along with biophysical modeling, made the form of the ER accessible to analysis. The factors that govern ER structure are now much better understood. This is particularly true of sheets versus tubules, the three way tubule junctions and to some extent, the junction of ER tubules coming out of the edge of a sheet. However, with all this activity, the subject of the connection of the ER to the nucleus has not been examined in detail. What makes it different is that the tubule is connected perpendicular to the plane of a sheet.

We thank the reviewer for appreciating the quality and novelty of our work.

*The manuscript uses the best ultrastructural techniques and provides strong evidence for a narrow neck at this connection in HeLa cells. With the same methodology, yeast cells (*S. cerevisiae*) have a wider connection. OpenOrganelle data from other mammalian cell types was examined. This data has less resolution and although it does not contradict the HeLa cell data, it does not support it strongly. As mentioned in the response to one of this reviewer's comments above, the spatial resolution of FIB-SEM datasets is good enough to examine if ER-NE junctions are narrower than ER-ER junctions. We think that our observation of several mammalian cells in FIB-SEM datasets strongly supports the conclusion that ER-NE junctions are narrower than ER-ER junctions and extends our findings in HeLa cells to two other mammalian cell types.*

This work is of interest to cell biologists specializing in membranous organelles or those interested in nuclear physiology. The connection of ER to nuclear envelope is an interesting problem that has not been studied recently. This manuscript could very well serve as a starting point for further structural or functional work by the authors or other groups.

We thank the reviewer for appreciating the significance and impact of our work.

Reviewer #4 (Evidence, reproducibility and clarity (Required)):

Summary:

Membrane bound ribosomes and ER exit sites are present in the cytosolic side of nuclear envelope (NE), suggesting that NE shares protein translocation, folding and quality control functions with the

endoplasmic reticulum (ER). Moreover, membrane continuity between the ER and outer NE membrane is evident, and, thus, NE is considered as a subdomain of the ER. To support this, during cell division, NE loses its identity, and participates to daughter cells as part of the ER. However, NE has also membrane proteins and luminal proteins that are enriched to NE and absent from ER during interface, and the segregation of NE specific proteins/lipids occurs concomitantly with NE formation during late anaphase/telophase. In this study, the ultrastructure of the ER-NE junctions is described using high resolution electron tomography. Results show convincingly a specific constriction at the ER-NE neck during interface in several mammalian cell types. This structure is absent during metaphase, and also from the budding yeast. Authors present a model for the formation of ER-NE junctions in higher eukaryotes and speculate about their functional role.

We thank the reviewer for the appreciation of our work and the valuable suggestions for further improvements.

Major comments:

The main conclusion of the paper is that although the ER and outer NE membranes are continuous, a specific hourglass shaped constriction at the neck is found in higher mammalian cells during interphase. The structure is specific to ER-NE necks, as it is absent during metaphase and ER-ER junctions. For the analysis, authors used high pressure freezing to ensure best structural preservation. Unfortunately, fixation is not the only potential source of artifacts; during tomography at ambient temperature, the thinning of the plastic sections under the beam can be up to 30%. In evaluation of the results, authors should consider how this thinning could affect the measurements of membrane distances and luminal width, and what type of distortions may happen as a consequence of asymmetric shrinkage.

We agree with the reviewer that it is important to clarify the degree of shrinkage and deformation of the sample that our EM protocol might introduce. Indeed, the irradiation of electrons shrinks the resin-embedded cells in the sections. To evaluate the degree of sample shrinkage, we compared the diameter of the nuclear pore complex, a regular macromolecular complex spanning the double membrane of the NE, in our EM tomograms with the one determined by cryo-EM tomography. The nuclear pore diameter in our EM tomograms of interphase HeLa cells was $77 \text{ nm} \pm 3.2 \text{ nm}$ (mean \pm S.D., 80 pores in 9 cells, new Figure EV3A,B), while it is on average 92 nm in cryo-EM tomograms of interphase HeLa cells that were vitrified by plunge freezing (Mosalaganti et al., *Science*, 2022, DOI: 10.1126/science.abm9506). Thus, the shrinkage of the specimen in our EM tomograms is 17%, which is consistent with the reported values of the shrinkage in epoxy resin (e.g. Kizilyaprak et al., *J. Struct. Biol.*, 2015, DOI: 10.1016/j.jsb.2014.10.009; Otsuka et al., *Elife*, 2016, DOI: 10.7554/eLife.19071.001). This means that our measurements on the junction width are 17% smaller than the native value. We have included this analysis (new Figure EV3), and clarified the shrinkage value in the main text on page 7.

To minimize the variation in shrinkage in different cells and to ensure a homogenous shrinkage throughout individual cells, we pre-irradiated the cells with the electron beam at a low magnification on the entire cell. To verify that the variation of the shrinkage between different samples is minimal, we compared the nuclear pore diameter in our EM tomograms. The nuclear pore diameter was comparable between different cells (new Figure EV3C), and thus we think that the shrinkage occurred to a similar degree in our EM tomograms and does not significantly affect our structural analysis. We have added this analysis and clarified how we minimized the shrinkage variation in the Method section on page 20.

In addition to analysis of own samples, authors took advantage of the publicly available whole-cell datasets in OpenOrganelle and used these datasets to expand the number of cell types analyzed. Moreover, the 3D-datasets were generated with different imaging technique, FIB-SEM. Although this technique provides lower resolution in general, it provides isotropic resolution, and the data could be used to eliminate the shortcomings of the tomography, thinning of the sections and the missing wedge. The authors could expand the comparison of the data from these different sources from this

perspective, especially since HeLa cells were used in their own tomography studies and FIB-SEM datasets in OpenOrganelle. Similarly, it would be interesting to see if similar approach could be used to compare their results to those obtained by cryo-EM by utilizing the cryo-EM database. Have authors checked if any suitable datasets for analysis of ER-NE junctions could be found from public archives?

The spatial resolution of the FIB-SEM datasets is good enough to examine if ER-NE junctions are narrower than ER-ER junctions as shown in Figure 5A–E, but is not high enough to inspect the exact morphology of the membrane constriction. Therefore, we would refrain from extracting fine details of the ultrastructure of ER-NE junctions from FIB-SEM. We would also want to emphasize that the missing wedge effect is minor in our EM tomograms, because we took dual axis tomography that minimizes this missing wedge effect (Arslan et al., *Ultramicroscopy*, 2006, DOI: 10.1016/j.ultramic.2006.05.010; Winter & Chlanda, *J. Struct. Biol.*, 2021, DOI: 10.1016/j.jsb.2021.107742) for most of the junctions. Since we did not highlight this point in our previous manuscript, we have now clarified this point in the Method section on page 21. Regarding the effect of sample shrinkage in our EM tomograms, we have defined its effect in the main text as described in the response to the above comment.

We agree that it would be good to compare the ultrastructure of ER-NE junctions in our EM tomograms with the one in cryo-EM. We looked for ER-NE junctions in cryo-EM tomograms of HeLa and other mammalian cells that are publicly available on EMPIAR (the Electron Microscopy Public Image Archive). Unfortunately, we could not find ER-NE junctions in those datasets. Given that the frequency of ER-NE junctions is very low (~0.13 junction per square micrometer, Figure 4C), more cryo-EM tomograms of the NE would need to be taken and inspected to find them in the future.

For the analysis of mitotic cells, double thymidine block was used to synchronize the cell culture. It is not clear, why synchronization was necessary, as CLEM was used to select the cells, and their number was rather low. Do cells continue growing and synthesizing new proteins during thymidine blocks? As one way to control potential artifacts due to the synchronization treatment, authors could compare the average thickness of ER and NE in naturally occurring interphase and mitotic cells vs. synchronized cells.

The reason why we synchronized the cell cycle is to enrich cells in late anaphase and early telophase in the same plastic sections, so that we can compare the ultrastructure side-by-side among several dividing and interphase cells under the electron microscope. To assess if the double thymidine block affects the ER and NE morphology, we compared the thickness of the ER and NE in the synchronised and non-synchronised cells, as the reviewer kindly suggested. As shown in the plot, the NE thickness is comparable between synchronised and non-synchronised HeLa cells in published EM datasets (Otsuka et al., *Elife*, 2016, Figure 2C, DOI: 10.7554/eLife.19071; Otsuka et al., *Nat. Struct. Mol. Biol.*, Supplementary Figure 2b, DOI: 10.1038/s41594-017-0001-9). In addition, the ER junction width is also comparable between synchronised (Figure 2I) and non-synchronised (Figure 5E, HeLa) cells (p -value > 0.1, unpaired two-tailed t-test). These lines of evidence indicate that the double thymidine block does not alter the width of the ER and the NE, and thus does likely not affect the ER-NE junction morphology. Of note, we observed cells around 10 hours after the release from the second thymidine block, and the effect of the thymidine block, if any, would be mitigated during the 10 hour incubation in the normal medium. We clarified these points in the main text and Method on pages 5 and 18.

Minor comments:

On page 5, last chapter (+ Fig.1 legend and materials and methods): "the quick tomograms covered the entire NE" is misleading, as the imaging covered a thin layer of the entire NE only. - Authors

could have analyzed the entire NE from the FIB-SEM datasets but chose to use stereological approach to minimize their work.

We have modified the text to make it clear that the quick tomograms covered the NE in a section and not the entire NE of the cell in the main text on page 6, and the legend of Figure 1.

To save time from the readers to follow the reference, authors could describe how the specimens used in OpenOrganelle datasets were fixed and processed, especially as they emphasize the importance of high pressure freezing in their own sample prep. Similarly, in Fig.4 legend, authors refer to measurements done in the previous study without explaining how and from what type of data.

We thank the reviewer for pointing these out. We have described how the OpenOrganelle datasets were generated and how the nuclear surface area measurement was done in the text on pages 10 and 11.

Is there a difference between mesh generation and segmentation, or is it just two different terms used for the same thing by different programs?

We apologize our short description of these terms. We have clarified these terms in the Method on page 24.

Reviewer #4 (Significance (Required)):

General assessment:

ER-NE gates were described earlier in the literature for specific cell types using standard thin-section TEM imaging, and in this study, the analysis was done with modern technology at 3D. The text is fluent and clear, and the quality of the images was excellent. The analysis of the data was thorough, and materials and methods including image analysis part were presented accurately and clearly. Ultrastructural analysis was done systematically, and generated models are beautiful and informative. Much thought has put into planning of the experiments and experimental approach. The shortcoming of the study is its limitation to ultrastructural analysis only without attempts to connect to any mechanism. The discussion part contains lot of speculation of the factors that might be needed for the formation and maintenance of the constriction and present several hypotheses for the function of the constriction. The paper would be much stronger if one of few of the leads would be followed, and if there would be any explanation for the role of these structures, or factors affecting them.

We thank the reviewer for the appreciation of the clarity and quality of our work. The molecular mechanism that regulates the function, shape and biogenesis of ER-NE junctions will be the subject of future studies, for which our discovery of a highly-constricted morphology of the ER-NE junctions lays the groundwork.

Advance:

The paper provides a very nice example for the reuse of publicly archived imaging datasets to complement own experimental work. Hopefully this paper encourages others to the same path, as the large volume EM datasets require significant investments and contain wealth of potential for reuse.

We strongly agree with the reviewer. The volume EM datasets that are publicly available contain wealth of potential for new discoveries. We also hope that our paper encourages other scientists to make good use of those datasets and also to deposit their own data to the public databases. We have deposited the tomograms containing junctions to the Electron Microscopy Public Image Archive (EMPIAR; <https://www.ebi.ac.uk/pdbe/emdb/empiar/>) under accession codes 12025, 12048, 12050, 12051, so that anyone can observe and judge the ultrastructure of the junctions. The tomograms will be available after the publication of this manuscript, but can be accessed for reviewers by these credentials: *information redacted*.

The paper strengthens the description of the ER-NE junction structure significantly and convincingly but does not further our understanding of the mechanisms behind the structure nor the function of them and raises more questions than provides answers. For structural analysis of this kind, the state-of-the-art technology is cryo-EM (e.g., preparation of lamella with cryo-FIB-SEM followed by cryo-tomography), and in this study, the technical limitations come from plastic embedding and ambient temperature imaging. The used techniques would be more adequate for cell biological study, where the described structure is somehow connected to the function in cell, or the factor(s) needed to the formation or maintenance are identified.

Indeed, a limitation of our current study is that we did not reveal the underlying molecular mechanism and the functions of the constricted morphology of ER-NE junctions. We do not think that cryo-EM is necessarily required because we have collected evidence that the ER-NE connections are distinct from the ER-ER junctions in not only our EM tomography data (Fig. 2) but also in the EM datasets deposited in public databases (Fig. 5).

Audience:

This study will be of special interest to cell biology community. The study could be an opening to several lines of research, e.g., identification of the factors forming or maintaining the structure, the potential function of the structure, how the structure affects the dynamics of the NE/ER membrane and luminal proteins.

We thank the reviewer for appreciating the impact of our work.

Reviewer's expertise:

The reviewer has long experience in electron microscopy, volumeEM techniques and image analysis, and operates mainly in the field of cell biology.

Dear Dr. Otsuka

Thank you for the submission of your revised manuscript to EMBO reports. We have now received the reports from the referees who were asked to evaluate it (copied below).

As you will see, both referees are very positive about the study and support publication without further revisions.

Browsing through the manuscript myself, I noticed a few editorial things that we need before we can proceed with the official acceptance of your study:

- Please provide up to 5 keywords.
- Please update the 'Conflict of interest' paragraph to our new 'Disclosure and competing interests statement'. For more information see <https://www.embopress.org/page/journal/14693178/authorguide#conflictsofinterest>
It also needs to be placed after the Acknowledgments section.
- Regarding the Author Contributions, we now use CRediT to specify the contributions of each author in the journal submission system. Therefore, please remove the Author Contributions from the manuscript file and make sure that the author contributions in our manuscript tracking system are correct and up-to-date. The information you specified in the system will be automatically retrieved and typeset into the article. You can enter additional information in the free text box provided, if you wish.
- Reference list: please only list the first 10 authors followed by et al.
- Please remove the DOI from Schindelin et al 2012 in the reference list. DOIs are only used for bioRxiv citations or Github/Zenodo citations (such as Gillies et al) but not for published manuscripts.
- All funding information must be entered in our online manuscript tracking system. This is the information that is transferred to our publisher and to PubMed. In this regard we note that the following are missing in the system: DOC Fellowship of the Austrian Academy of Sciences (no. 25951), Max Perutz PhD Fellowship (University of Vienna and the Medical University of Vienna).
- I note that you provided Figure EV4 as eps file in a zip folder since it is 304 MB. I am not sure that this can be typeset and used as a zip archive. Any chance to reduce file size, while maintaining good resolution?
- In text callouts for Figure 3C and Figure 5D are missing and the following callout needs to be updated to the correct one: "Supplementary Figure 2b"
- Movie legends need to be removed from the manuscript. Each should be saved as a readme.txt file and then zipped up together with its corresponding movie; movies should then be uploaded as one folder (Movie EV1, etc.) per movie.
- Materials and Methods should be Methods
- Data availability section: please provide links that resolve directly to the datasets, not just to the database as such.
- Our production/data editors have asked you to clarify several points in the figure legends (see below). Please incorporate these changes in the manuscript and return the revised file with tracked changes with your final manuscript submission:
 - A) Please note that a separate 'Data Information' section is required in the legends of figures 2a-b, f-g; 3 a(i-iv), d (i-iv); 5a-c, f. This means that you add e.g., "Data Information: (A, B, F, G) Scale bars: 20 nm".
 - B) Please note that the scale bar needs to be defined for figure EV 2a and for figure EV4.
 - C) Figure EV4: please add more description so that the figure can be understood without having to go back to Figure 3. Please define the arrowheads.
- Finally, EMBO Reports papers are accompanied online by
 - A) a short (1-2 sentences) summary of the findings and their significance,
 - B) 2-3 bullet points highlighting key results and
 - C) a schematic summary figure that provides a sketch of the major findings (not a data image).Please provide the summary figure as a separate file in PNG or JPG format at a size of 550x300-600 pixels (width x height). Please note that the size is rather small and that text needs to be readable at the final size. Please send us this information along with the revised manuscript.

- On a different note, I would like to alert you that EMBO Press offers a new format for a video-synopsis of work published with us, which essentially is a short, author-generated film explaining the core findings in hand drawings, and, as we believe, can be very useful to increase visibility of the work. This has proven to offer a nice opportunity for exposure i.p. for the first author(s) of the study. Please see the following link for representative examples and their integration into the article web page:

<https://www.embopress.org/doi/full/10.15252/embj.2019103932>

With kind regards,

Referee #2:

The authors have addressed all the concerns and suggestions previously raised by this reviewer. The manuscript is now suitable for publication in EMBO Reports without further revision.

Referee #4:

The paper is technically sound and thoroughly done and have addressed all concerns that I raised in my initial review. Although the paper does not further our understanding of the mechanisms behind the structure or the function of NE-ER junctions, it will serve as initial high-quality description of the morphology of the NE-ER junction for further studies.

The paper provides a very nice example for the reuse of publicly archived imaging datasets to complement own experimental work, and provides new material for open access and potential reuse.

Rev_Com_number: RC-2023-02194

New_manu_number: EMBOR-2023-58627V2

Corr_author: Otsuka

Title: The endoplasmic reticulum connects to the nucleus by constricted junctions that mature after mitosis

All editorial and formatting issues were resolved by the authors.

Dr. Shotaro Otsuka
Max Perutz Lab, Vienna
Dr. Bohr Gasse 9
Vienna, Vienna 1030
Austria

Dear Shotaro,

I am very pleased to accept your manuscript for publication in the next available issue of EMBO reports. Thank you for your contribution to our journal.

Kind regards,

Martina
